# Metabolic reprogramming from glycolysis to fatty acid uptake and beta-oxidation in platinum-resistant cancer cells

Yuying Tan [1,6], Junjie Li [2,6] ✉, Guangyuan Zhao [3,6], Kai-Chih Huang[1], Horacio Cardenas[3], Yinu Wang[3], Daniela Matei [3,4] ✉ & Ji-Xin Cheng [1,2,5] ✉

Increased glycolysis is considered as a hallmark of cancer. Yet, cancer cell metabolic reprograming during therapeutic resistance development is under-studied. Here, through high-throughput stimulated Raman scattering imaging and single cell analysis, we find that cisplatin-resistant cells exhibit increased fatty acids (FA) uptake, accompanied by decreased glucose uptake and lipogenesis, indicating reprogramming from glucose to FA dependent anabolic and energy metabolism. A metabolic index incorporating glucose derived anabolism and FA uptake correlates linearly to the level of cisplatin resistance in ovarian cancer (OC) cell lines and primary cells. The increased FA uptake facilitates cancer cell survival under cisplatin-induced oxidative stress by enhancing beta-oxidation. Consequently, blocking beta-oxidation by a small molecule inhibitor combined with cisplatin or carboplatin synergistically suppresses OC proliferation in vitro and growth of patient-derived xenografts in vivo. Collectively, these findings support a rapid detection method of cisplatin-resistance at single cell level and a strategy for treating cisplatin-resistant tumors.

Metabolic reprogramming in cancer cells has been recognized since the discovery of the Warburg effect in 1920s[1,2]. Increased aerobic glycolysis is now widely considered as a hallmark of many cancers and clinically exploited as a target for cancer therapy and a cancer biomarker for diagnosis[3]. In the past decade, numerous studies have investigated the heterogeneity and complexity of cancer metabolism beyond the Warburg effect[4]. Metabolic reprogramming allows cancer cells to adapt to intrinsic or extrinsic cues from the microenvironment through plasticity and high flexibility in nutrient acquisition and utilization[5]. Particular attention has been paid to metabolic alterations associated with critical steps of cancer progression, such as metastasis initiation, circulation, and colonization[5–7]. Metabolic reprogramming in cancer stem cells identified potential vulnerabilities for cancer stem cells targeting therapy[8,9]. Cancer cells also rewire their metabolic

dependencies within a specific microenvironment niche by interacting with stroma cells[10] or with the surrounding adipocytes[11,12]. Further, alterations in nutrient utilization under metabolic stress conditions have recently been reported[13–15]. Despite these recent advances, the understanding of cancer cell metabolism remains incomplete. One of the less studied areas is cancer metabolic reprogramming associated with resistance to therapy.

Therapeutic resistance remains one of the biggest challenges facing cancer treatment. Resistance to chemotherapy or molecularly targeted therapies is a major cause of tumor relapse and death[16]. Emerging studies support an association between metabolic reprogramming and cancer drug resistance[17,18]. Several studies have linked the Warburg effect to resistance to radiation[19] and lactate production was shown to promote resistance to chemotherapy in cervical

[1]Biomedical Engineering, Boston University, Boston, MA 02155, USA. [2]Electrical and Computer Engineering, Boston University, Boston, MA 02155, USA. [3]Department of Obstetrics and Gynecology, Feinberg School of Medicine, Northwestern University, Chicago, IL 60611, USA. [4]Robert H. Lurie Comprehensive Cancer Center, Chicago, IL 60611, USA. [5]Photonics Center, Boston University, Boston, MA 02155, USA. [6]These authors contributed equally: Yuying Tan, Junjie Li, Guangyuan Zhao. ✉e-mail: junjiel168@gmail.com; daniela.matei@northwestern.edu; jxcheng@bu.edu

cancer[20]. Altered lipid metabolism has also been implicated in the acquisition of drug resistance[21]. Increased de novo lipogenesis mediated by FASN facilitated gemcitabine resistance in pancreatic cancer[22] while cancer-associated adipose tissue promoted resistance to anti-angiogenic interventions by supplying FA to cancer cells in regions where the glucose demand was insufficient[23]. Additionally, lipid droplet production mediated by lysophosphatidylcholine acyltransferase 2 promoted resistance of colorectal cancer cells to 5-fluorouracil and oxaliplatin[24]. Recently, it has been proposed that drug-tolerant cells adopt a state of diapause similar to suspended embryonic development to survive chemotherapy toxic insults, in which cell proliferation and metabolic processes are suppressed[25]. These studies support that metabolic reprograming underlie the development of drug resistance and point to potential metabolic vulnerabilities of resistant cancer cells, which remain underutilized.

Platinum-based drugs, including cisplatin, carboplatin and oxaliplatin, represent one class of the most widely used chemotherapy drugs[26]. Resistance to platinum is a barrier to effective treatment in multiple cancers, including ovarian, testicular, bladder, head and neck, non-small-cell lung cancer and others[27]. Understanding the metabolic reprograming underlying platinum-resistant cancer cells is critical for the development of effective treatment strategies. Yet, precisely profiling metabolic reprogramming using conventional technology is difficult, because within a cell population, only a small portion of cells is drug resistant or tolerant. In this study, by taking advantage of a hyperspectral stimulated Raman scattering (SRS) imaging platform, we depict the metabolic profile of platinum-resistant cancer cells at the single cell level.

SRS microscopy is a recently developed label-free chemical imaging technique that detects the intrinsic chemical bond vibrations[28–31]. The value of SRS microscopy was demonstrated in identifying cholesteryl ester accumulation as a signature associated with multiple aggressive cancers[32,33], discovering increased lipid desaturation in OC stem cells[8], and tracing metabolic flux by isotope labeling[34–36]. More recently, large-area hyperspectral SRS microscopy and high-throughput single-cell analysis revealed lipid-rich protrusion in cancer cells under stress[37]. Raman spectro-microscopy-based single cell metabolomics unveiled an important role of lipid unsaturation in aggressive melanoma[38].

Here, through large-area hyperspectral SRS imaging and subsequent single-cell analysis, we identified a stable metabolic switch from glucose and glycolysis dependent to FA uptake and fatty acid beta-oxidation (FAO) dependent anabolic and energy metabolism in cisplatin-resistant OC cells. By coupling metabolic flux through isotope labeling and SRS-based molecular imaging, we found that cisplatin-resistant cells display increased uptake of exogenous FA, accompanied with decreased glucose uptake and de novo lipogenesis. By incorporating SRS imaging-based measurements of glucose-derived anabolism and FA uptake, we introduce the "metabolic index" defined as the ratio of FA uptake versus glucose incorporation. The metabolic index was found to linearly correlate to the level of resistance to cisplatin in OC cell lines and in primary human cells, demonstrating the potential of using SRS imaging for rapid detection of cisplatin-resistance ex vivo. Mechanistically, cisplatin-resistant cells display higher FAO rate, which supplies additional energy and promotes cancer cell survival under cisplatin-induced oxidative stress. Blocking FAO by a small molecule inhibitor or genetic perturbation in combination with cisplatin or carboplatin treatment synergistically suppressed OC proliferation in vitro and the growth of a patient-derived xenograft model in vivo. In addition, acute treatment with cisplatin induced a transient metabolic shift towards higher FA uptake in lung, breast, and pancreatic cancers. Together, these results promise treatment options for patients with cisplatin-resistant tumors by targeting the FAO pathway.

## Results

### High-throughput SRS imaging unveils lipid accumulation in OC cells with platinum resistance

To identify the altered lipid metabolism in cisplatin-resistant cells, we established a high-throughput single-cell analysis approach, which couples large-area hyperspectral SRS scanning of 200–500 cells per group with spectral phasor segmentation and CellProfiler analysis. As shown in Supplementary Fig. 1a, we firstly acquire a stack of large-area hyperspectral SRS images, containing hundreds of individual cells in each field of view. An SRS spectrum is extracted at each pixel from the image stack. Then, the hyperspectral SRS images are segmented through a spectral phasor algorithm to generate maps of intracellular compartments corresponding to nuclei and lipids (mostly in lipid droplets) based on the spectrum similarity[39]. Next, the nuclei map is inputted into CellProfiler[39] to guide the identification of the edges of each individual cell from the raw whole cell image. After individual cells are outlined, the lipid map is mapped back to the corresponding cells. Finally, quantitative characterization of lipids in terms of integrated intensity, mean intensity, area, and lipid droplet size in each individual cell is performed.

To explore the lipid metabolic signature of cisplatin-resistant OC cells, we generated isogenic pairs of cisplatin-resistant cells from three OC cell lines, including SKOV3, OVCAR5, and COV362, through repeated long-time exposures and recoveries after cisplatin treatment at IC50 concentration[40]. Resistance to cisplatin in these cell lines was validated by repeat assays measuring cisplatin dose response. All the resistant cell lines exhibited 2-3-fold increase of $IC_{50}$ when compared to their parental counterpart cell lines (Supplementary Fig. 1b-e). In addition, we studied isogenic PEO1/PEO4 cell lines derived from the same patient, at the time of a platinum-sensitive (PEO1) and platinum-resistant-recurrence (PEO4)[41].

Taking advantage of the high-throughput imaging analysis platform, we analyzed the lipid metabolism in these 4 pairs of cisplatin-resistant and parental OC cell lines. Comparing SRS images of sensitive PEO1 and cisplatin-resistant PEO4 cells, there tended to be an increase of lipid intensity in PEO4 cells but with large cell-to-cell variations (Fig. 1a). We quantitatively analyzed the integrated lipid intensity in individual cells and plotted them in histograms. Intriguingly, the histograms displayed two distinct subpopulations, lipid-poor and lipid-rich, in each cell line, implying the existence of metabolic heterogeneity within the same group. While lipid-poor cells dominate in PEO1 cell line, PEO4 cells show a dramatic increase in lipid-rich subpopulation and a decrease in lipid-poor subpopulation (Fig. 1b). Single-cell analysis reveals an even more obvious increase in lipid-rich subpopulation and decrease in lipid-poor subpopulations in SKOV3-cisR cells, compared to SKOV3 (Fig. 1c, d). Additionally, we observed similar lipid content changes in the other two pairs of cell lines, OVCAR5 versus OVCAR5-cisR (Supplementary Fig. 1f), and COV362 versus COV362-cisR (Supplementary Fig. 1g), supporting that cisplatin-resistant cells harbor higher levels of lipid accumulation. Furthermore, after acute treatment with cisplatin, a significant increase in lipid-rich subpopulation and decrease of lipid-poor subpopulation was found in SKOV3 cells (Fig. 1e), but no obvious change of lipid distribution pattern in SKOV3-cisR cells were detected (Fig. 1f), supporting that lipid-rich cells are more resistant to cisplatin treatment. To determine whether lipid accumulation also occurs in vivo in platinum-treated tumors, we performed SRS imaging of lipids in OVCAR5 xenografts collected from mice treated weekly with saline or carboplatin for 3 weeks. As anticipated, tumor growth was suppressed by carboplatin treatment (Supplementary Fig. 1h). However, cells isolated from xenografts residual after carboplatin treatment showed increased resistance to carboplatin in in vitro treatment, compared to cells isolated from the saline-treated tumors. (Supplementary Fig. 1i). The results shown in Fig. 1g, h indicate heterogenous lipid

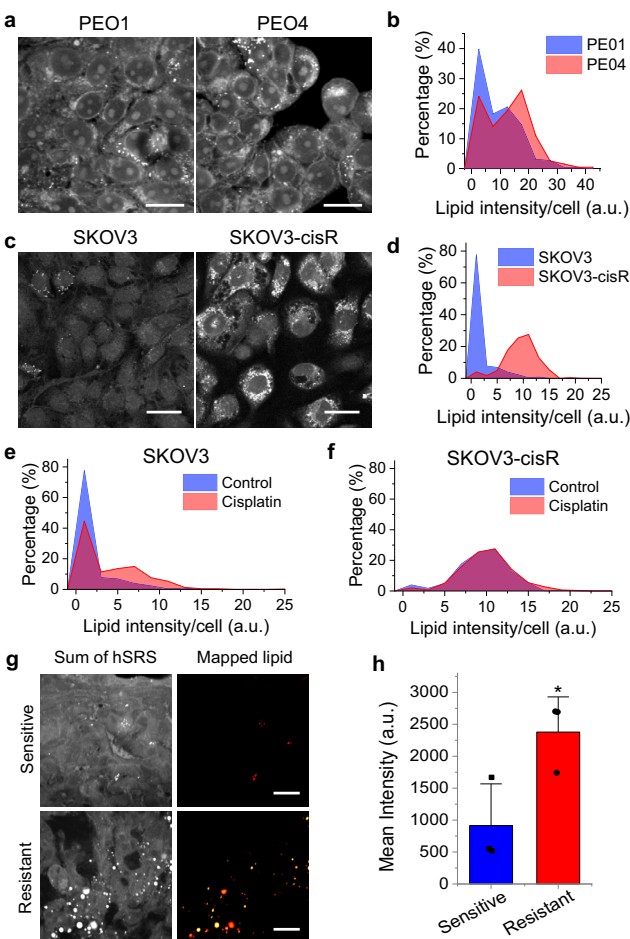

**Fig. 1 | High-throughput imaging of lipid metabolism in isogenic pairs of cisplatin-sensitive and -resistant OC cells. a** Representative large-area SRS images of parental PEO1 and cisplatin-resistant PEO4 cells. **b** Histograms of integrated cellular lipid intensity in PEO1 and PEO4 cells generated through high-throughput single-cell analysis. **c** Representative large-area SRS images of parental SKOV3 and cisplatin-resistant SKOV3-cisR cells. **d** Histograms of integrated cellular lipid intensity in SKOV3 and SKOV3-cisR cells. **e** Histograms of integrated cellular lipid intensity in SKOV3 cells treated with or without cisplatin. **f** Histograms of integrated cellular lipid intensity in SKOV3-cisR cells treated with or without cisplatin. **g** Representative hyperspectral SRS image (sum of all channels) and Phasor mapped lipid image of sliced OVCAR5 xenograft tumor tissue from mouse treated with vehicle (sensitive) or carboplatin (resistant). **h** Quantitative analysis of SRS signal from lipid in carboplatin sensitive and resistant ovarian tumor tissue by mean intensity. Data are presented as means + SD; $n = 3$ animals; two-sided Student's $t$ test; $P = 0.043$; *$P < 0.05$. Scale bar: 20 μm. Source data are provided in the Source Data file.

accumulation and higher lipid amount in the carboplatin-treated tumors compared with the platinum-sensitive tumors. These data collectively suggest that higher level of lipid content is a metabolic feature of cisplatin-resistant OC cells.

## Increased FA uptake but not de novo lipogenesis contributes to high-level lipid content in cisplatin-resistant OC cells

To identify the source of increased lipid content in cisplatin-resistant OC cells, we examined the contribution of de novo lipogenesis and of FA uptake, respectively. Using a stable isotope probing method[35], we examined the level of lipogenesis by feeding the cells with deuterium-labelled glucose-d7. Newly synthesized macromolecules (mostly lipids) were imaged by hyperspectral SRS microscopy at Raman shift from 2050 cm⁻¹ to 2350 cm⁻¹, covering the vibrational frequencies of C-D

bonds. SRS images show weaker C-D signal in cisplatin-resistant PEO4 cells than the signal in parental PEO1 cells (Fig. 2a). Quantitative analysis confirmed significant reduction of both signal intensity and relative area fraction in PEO4 cells when compared to PEO1 cells (Fig. 2b), indicating a decrease in glucose-derived anabolism and de novo lipogenesis in cisplatin-resistant cells. Using a similar approach, we examined the FA uptake by hyperspectral SRS imaging at C-D vibrational frequencies in cells fed with deuterium-labelled palmitic acid-d31 (PA-d31). In contrast to glucose-d7 fed cells, C-D signal in PA-d31 fed PEO4 cells was stronger than PEO1 cells (Fig. 2c). Quantitative analysis revealed significant increase of both signal intensity and relative area fraction (Fig. 2d). In addition to saturated FA (PA-d31), we further tested the uptake of an unsaturated FA, oleic acid-d34 (OA-d34). OA-d34 uptake was significantly increased in PEO4 cells comparing to PEO1 cells (Fig. 2e, f). These results suggest the increased uptake of FA is not specific to certain type of FA, but rather reflects a general upregulation of FA uptake pathway.

To verify if the observed phenomenon is cell type specific, we repeated the measurements in SKOV3 and SKOV3-cisR cells. Consistently, SRS images and quantitative analysis showed a significant decrease in glucose-d7 derived C-D signal in SKOV3-cisR cells (Fig. 2g) and increase in PA-d31 signal (Fig. 2h) and OA-d34 signal in SKOV3-cisR cells (Fig. 2i), when compared to parental SKOV3 cells. Additionally, we observed the same trend in the other two pairs of cell lines, OVCAR5 versus OVCAR5-cisR (Supplementary Fig. 2a) and COV362 versus COV362-cisR (Supplementary Fig. 2b). In addition, inhibition of de novo lipogenesis by the FASN inhibitor C-75 did not affect the increased lipid content in SKOV3-cisR compared with SKOV3, indicating that the enhanced lipid amount in cisplatin-resistant cells is independent of de novo lipogenesis (Supplementary Fig. 2c, d). These data collectively suggest a metabolic switch from glucose-derived anabolism to FA uptake in cisplatin-resistant OC cells.

## Metabolic index as a predictor of cisplatin resistance

Having shown decreased glucose-derived anabolism and increased FA uptake in cisplatin-resistant OC cells, we continued exploring whether this metabolic feature can be used for differentiation of cisplatin-resistant from cisplatin-sensitive cancer cells. To quantitatively characterize resistance to cisplatin, we calculated IC50 dose of cisplatin in various cell lines and their C-D intensities (presented as area fraction) from glucose-d7, PA-d31, or OA-d34 (Supplementary Table 1). Interestingly, glucose-d7 derived C-D intensity was found negatively correlated to IC50 to cisplatin (Fig. 3a), while PA-d31 intensity was positively correlated to IC50 to cisplatin (Fig. 3b). To integrate two measurements into one, we used the ratio of PA-d31/(PA-d31 + Glucose-d7), giving a dimensionless number ranging from 0 to 1. We defined this ratio as the "metabolic index". The index linearly correlated to the IC50 to cisplatin (Fig. 3c), suggesting its potential to detect and quantitatively determine resistance to cisplatin in cancer cells.

Understanding the potential value of this metabolic imaging method for detecting cisplatin resistance at the single cell level, we further applied hyperspectral SRS imaging to simultaneously measure glucose-derived anabolism and FA uptake in the same cells cultured with Raman probes of FA and glucose. Specifically, we used glucose-d7 to follow glucose anabolism[35,36]. Instead of using deuterium-labelled FA to trace FA uptake, we used a FA analog, 17-octadecynoic acid (ODYA). ODYA has an endogenous C≡C at one end of the FA chain, which produces a strong Raman peak around 2100 cm⁻¹ (Supplementary Fig. 3a). The distinctive Raman spectrum of ODYA enables spectral separation of C≡C labeled FA (from FA uptake) from C-D labeled macromolecules derived from glucose-d7 (Supplementary Fig. 3a). To test this in a biological environment, we performed hyperspectral SRS imaging in cells fed with both ODYA and glucose-d7. From images, we observed two signals with distinctive spectra, from C≡C labeled FA and C-D derived from glucose-d7, respectively (Supplementary Fig. 3b).

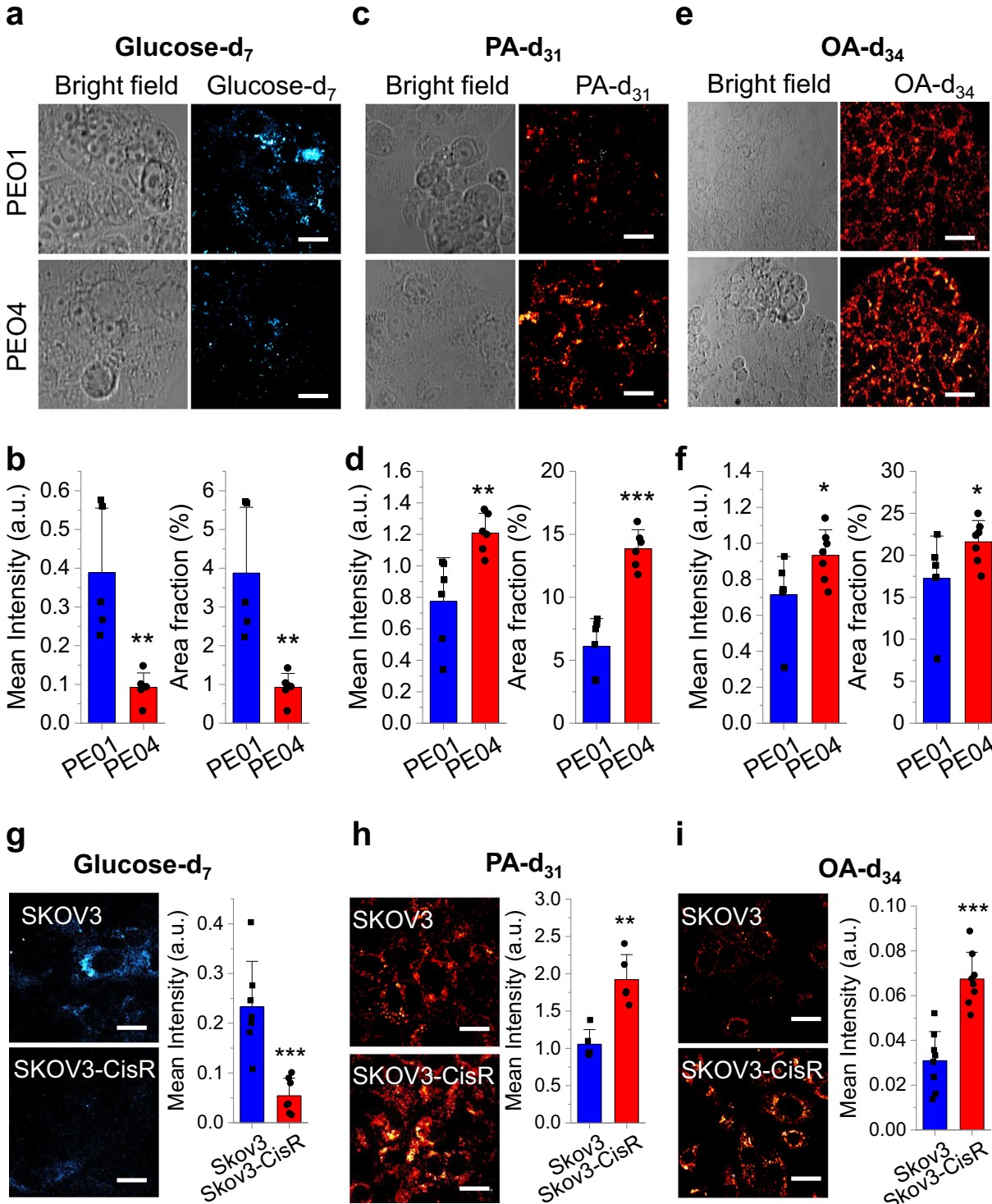

**Fig. 2 | Increased FA uptake, not de novo lipogenesis, is the major contributor to lipid accumulation in cisplatin-resistant OC cells. a** Representative bright field and SRS images of PEO1 and PEO4 cells fed with glucose-d$_7$ for 3 days.
**b** Quantitative analysis of SRS signal of C-D bonds in glucose-d$_7$ fed PEO1 and PEO4 cells by mean intensity and area fraction. $n = 5$. $P = 0.0076$ and $0.0083$.
**c** Representative bright field and SRS images of PEO1 and PEO4 cells fed with PA-d$_{31}$ for 6 h. **d** Quantitative analysis of SRS signal of C-D bonds in PA-d$_{31}$ fed PEO1 and PEO4 cells by mean intensity and area fraction. $n = 6$. $P = 0.0051$ and $3 \times 10^{-5}$.
**e** Representative bright field and SRS images of PEO1 and PEO4 cells fed with OA-d$_{34}$ for 6 h. **f** Quantitative analysis of SRS signal of C-D bonds in OA-d$_{34}$ fed PEO1 ($n = 6$) and PEO4 ($n = 7$) cells by mean intensity and area fraction. $P = 0.030$ and $0.0048$.

**g** Representative SRS images of SKOV3 and SKOV3-cisR cells fed with glucose-d$_7$ for 3 days and quantitative analysis of SRS signal of C-D bonds by mean intensity. $n = 7$. $P = 0.00075$. **h** Representative SRS images of SKOV3 and SKOV3-cisR cells fed with PA-d$_{31}$ for 6 h and quantitative analysis of SRS signal of C-D bonds by mean intensity. $n = 5$. $P = 0.0010$. **i** Representative SRS images of SKOV3 and SKOV3-cisR cells fed with OA-d$_{34}$ for 6 h and quantitative analysis of SRS signal of C-D bonds by mean intensity. $n = 8$. $P = 2.2 \times 10^{-5}$. Data in all the bar charts (**b**, **d**, **f** and **g–i**) are shown as means + SD. All n represents technical replicates. Statistical significance was analyzed using one-sided Student's t test. *$P < 0.05$, **$P < 0.01$, and ***$P < 0.001$. Scale bar: 20 μm. Source data are provided in the Source Data file.

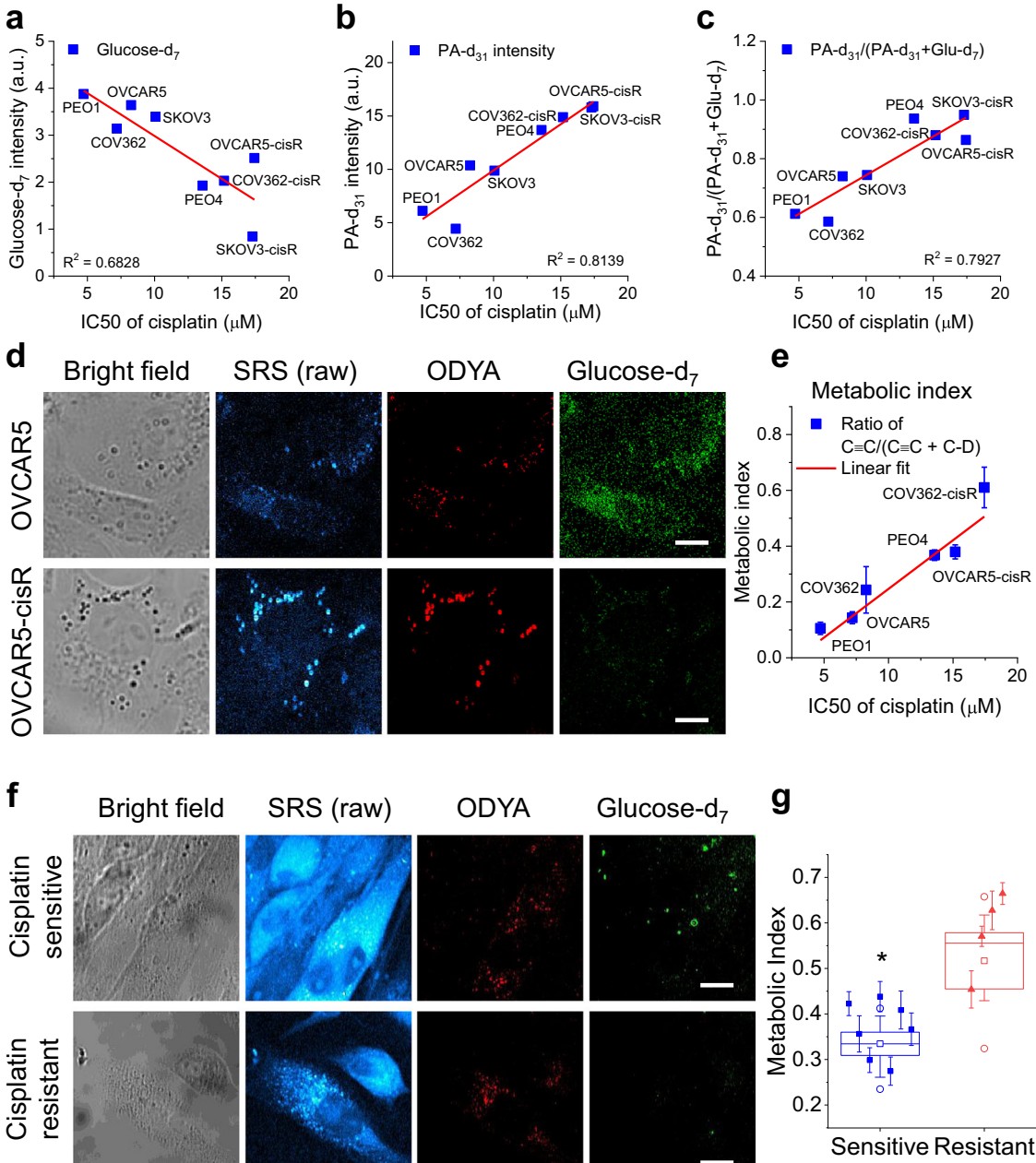

**Fig. 3 | Metabolic index by integrating glucose-derived lipogenesis and FA uptake directly correlates with cisplatin resistance. a** A linear regression of glucose-d$_7$ intensity to IC$_{50}$s of cisplatin in various OC cell lines. **b** A linear regression of PA-d$_{31}$ intensity to IC$_{50}$s of cisplatin in various OC cell lines. **c** A linear regression of the ratio of PA-d$_{31}$/(PA-d$_{31}$ + Glucose-d$_7$) to IC$_{50}$s of cisplatin in various OC cell lines. $n = 6$ technical replicates for (**a**–**c**). **d** Representative bright field images, raw SRS images, and processed SRS images of ODYA and glucose-d$_7$ in OVCAR5 and -cisR cells. **e** A linear regression of the metabolic index, as defined by the ratio of C≡C/(C≡C + C-D) to IC$_{50}$s of cisplatin in various OC cell line pairs (COV362 ($n = 8$), PEO ($n = 6$ and 7) and OVCAR5 ($n = 4$)). n represents technical replicates. $R^2 = 0.9235$. Data is shown as mean ± SEM. **f** Representative bright field images, raw SRS images, and processed SRS images of ODYA and glucose-d$_7$ in primary OC cells from cisplatin treatment-resistant or sensitive patient. **g** Quantitative analysis of metabolic index (the ratio of C ≡ C/(C ≡ C + C-D) for primary OC cells from cisplatin treatment-resistant or sensitive patient. Each data point represents the average metabolic index of individual cancer cells from a patient and its error bar indicates the SEM; $n = 30, 31, 19, 25, 27, 33, 12, 11, 24, 20$ and 30 cells. The box plot indicates the analysis for each group (sensitive ($n = 7$ biological replicates) vs. resistant ($n = 4$ biological replicates)). The bound of outer box, inner box, lines, whiskers, circles represent SEM, mean, medium, 25% to 75% of data, maxima and minima, respectively. Statistical significance used two-sided Student's $t$ test; $P = 0.011$. *$P < 0.05$. Scale bar: 20 μm. Source data are provided in the Source Data file.

Next, we applied this approach to image OVCAR5 and OVCAR5-cisR cells. Two components, C≡C and C-D, were segmented from the raw SRS images based on the spectral phasor algorithm[39]. Consistently, we observed apparently stronger C≡C signal and weaker C-D signal in OVCAR5-cisR cells, when compared to OVCAR5 cells (Fig. 3d). Quantitative analysis confirms a significant increase of C≡C and decrease of C-D signal. The metabolic index, ratio of C≡C/ (C≡C + C-D), is more significantly increased in OVCAR5-cisR cells (Supplementary Fig. 3c). Following this validated protocol, we analyzed metabolic indices in the other two pairs of cell lines, including PEO1 and PEO4 (Supplementary Fig. 3d, e), and COV362 and COV362-cisR cells (Supplementary Fig. 3f, g). Consistently, a linear correlation was established between metabolic index and IC$_{50}$s of cisplatin in these cell lines (Fig. 3e).

To further validate the metabolic index as a predictor of platinum resistance in clinically relevant samples, we applied this method to primary OC cells obtained from de-identified consenting patients for whom data on resistance/response to platinum was available. Patients' characteristics are included in Supplementary Table 2. As shown in Fig. 3f, in OC cells isolated from a patient with platinum-sensitive tumor, the signal from ODYA was observed only in some of the cells, but the signal from glucose-d$_7$ was relatively strong. In OC cells isolated from cisplatin-resistant tumors, ODYA signal was more evenly distributed in the cells imaged, while the signal from glucose-d$_7$ was weaker. Quantitative analysis showed that the metabolic indices were higher in samples from four patients with resistant tumors, when compared to cancer cells from seven patients with the sensitive disease (Fig. 3g). The histogram of metabolic index data indicated a clear separation between the sensitive and resistant groups (Supplementary Fig. 3h). Receiver operating characteristic (ROC) analysis yielded a threshold value at 0.412 with high sensitivity of 1, specificity of 1 and AUC (area under curve) of 1, suggesting that the metabolic index has a high chance to successfully distinguish platinum-sensitive and resistant OC cells (Supplementary Fig. 3i). While studies from a larger pool of patients are needed to validate the applicability of the metabolic index measurement to detect platinum resistance, our pilot study suggests the potential clinical applicability of metabolic imaging for predicting response/resistance to platinum.

## FA uptake contributes to cisplatin resistance

Knowing that cisplatin-resistant OC cells uptake more FA, we asked whether the FA uptake is a cause or result of cisplatin resistance. Firstly, we tested whether modulating exogenous FA availability affect endogenous lipid amount in cisplatin-resistant OC cells. We cultured OVCAR5-cisR cells in lipid-deficient culture medium or normal medium supplemented with 1% lipid mixture for 24 h and then examined lipid amount by SRS microscopy. Lipid deficiency significantly reduced intracellular lipid amount while lipid supplementation increased the intracellular lipid amount (Fig. 4a, b). Similar phenomenon was observed in the SKOV3-cisR cell line (Supplementary Fig. 4a, b). These observations further support the forementioned conclusion that FA uptake, instead of de novo lipogenesis, is the major source of the lipid accumulation in cisplatin-resistant OC cells. Next, we examined whether modulating exogenous lipid availability impacts cancer cell's resistance to cisplatin. We found that lipid deficiency increased sensitivity to cisplatin, while lipid supplementation slightly decreased sensitivity to cisplatin in OVCAR5-cisR (Fig. 4c), PEO4 (Supplementary Fig. 4c), and SKOV3-cisR cells (Supplementary Fig. 4d). To check whether OC cells would upregulate glucose metabolism in a FA depleted environment, glucose uptake was measured in SKOV3 and SKOV3-cisR cells cultured with regular or lipid-deficient medium using fluorescent glucose analog 2-deoxy-2-[(7-nitro-2,1,3-benzoxadiazol-4-yl) amino]-D-glucose (2-NBDG). Glucose uptake remained similar in lipid sufficient and deficient environment (Supplementary Fig. 4e, f). These data suggest that resistance to cisplatin can be alleviated by modulating exogenous FA availability.

FA uptake is a cellular process facilitated by multiple FA transporters/carriers, including CD36, FATPs, and FABPs[42,43]. One of the key proteins reported to be upregulated in OC is the fatty acid binding protein 4 (FABP4)[12,44]. We assessed whether the increased FA uptake in cisplatin-resistant OC cells is regulated through upregulation of *FABP4*, but observed very low *FABP4* mRNA levels in OVCAR5 and OVCAR5-cisR cells, suggesting FABP4 is not likely to play a major role in mediating the increased FA uptake in cisplatin-resistant OVCAR5 cells. Next, we investigated the expression of a panel of other FA uptake regulator genes, including *CD36*, *FATP1-6*, *FABP5*, and *GOT2* (*FABP(PM)*)[45] and found expression of *FABP5* and *FABP(PM)* is higher than other genes (Supplementary Fig. 4g). Replication experiments confirmed a significant upregulation of *FABP5* and *FABP(PM)* in

resistant OVCAR5-cisR (Fig. 4d, e) and PEO4 cells (Supplementary Fig. 4h) compared to their parental cells, suggesting that FABP5 and FABP(PM) may mediate FA uptake in cisplatin-resistant cells. In addition, cisplatin treatment induced an acute rise of *FABP5* and *FABP(PM)* expression in cisplatin-sensitive cell OVCAR5 (Fig. 4f), further supporting the potential involvement of FABP5 and FABP(PM) in cisplatin resistance related FA uptake. In contrast, *mRNA* expression levels of glucose transporter *GLUT1* was reduced in resistant SKOV3-cisR compared to parental cells (Supplementary Fig. 4i). *GLUT1* expression was not significantly changed after cisplatin treatment in OVCAR5 cells, implying that *GLUT1* downregulation may be an adaptive change in cisplatin-resistant cells rather than an acute response to cisplatin treatment (Supplementary Fig. 4j). To rule out the possibility that the increased fatty acid uptake is caused by change in membrane fluidity at high concentrations of exogenous fatty acids, we performed SRS imaging of FA uptake at lower concentrations of PA-d$_{31}$ and OA-d$_{34}$ and found a similar trend of increased fatty acid uptake in resistant cells (Supplementary Fig. 4k-n). These data support that increased fatty update in cisplatin-resistant ovarian cancer cells is transporter mediated and likely due to an adaptive metabolic reprogramming in response to cisplatin treatment.

Next, we tested whether a potent inhibitor of FABP, BMS-309403 (BMS), can suppress FA uptake in cisplatin-resistant cells[46]. Treatment with BMS significantly reduced PA-d$_{31}$ uptake in SKOV3-cisR cells (Fig. 4g, h). Suppression of FA uptake by BMS was also observed in OVCAR5-cisR cells in a dose-dependent manner (Supplementary Fig. 5a, b). Furthermore, inhibition of FA uptake by BMS reduced resistance to cisplatin in multiple resistant cell lines, including PEO4 (Fig. 4i), SKOV3-cisR (Fig. 4j), and OVCAR5-cisR cells (Fig. 4k), while the effects of BMS were less obvious in the sensitive cell lines (Supplementary Fig. 5c-e). These results support deregulated FA uptake in the development of cisplatin resistance in OC cells.

## Increased FAO rate contributes to cisplatin resistance

Considering that one major function of lipids is energy production through FAO, we next investigated whether FAO is increased in cisplatin-resistant cancer cells. By measuring the oxygen consumption rate (OCR) in parental and resistant cancer cells, we found OVCAR5-cisR cells displayed much higher levels of oxygen consumption than OVCAR5 cells (Fig. 5a). To test whether the increased oxidation rate arises from FAO, we used etomoxir, an inhibitor of CPT1, which transports FA into mitochondria for FAO. Etomoxir did not induce an obvious change in oxygen consumption in OVCAR5 cells (Fig. 5b), but significantly reduced oxygen consumption in OVCAR5-cisR cells (Fig. 5c). Quantitation of OCR confirmed a significant reduction of OCR after etomoxir treatment in OVCAR5-cisR cells, but not in OVCAR5 cells (Supplementary Fig. 6a). FAO was also measured through the Seahorse FAO assay to assess etomoxir induced mitochondrial respiration change. As shown in Fig. 5d, OVCAR5-cisR cells had an overall higher OCR than the parental cells and showed an obvious reduction of OCR after treatment with etomoxir. In contrast, OCR of OVCAR5 cells was less sensitive to etomoxir treatment. Etomoxir-induced basal respiration, ATP production and maximal respiration reduction in resistant OVCAR5-cisR cells were significantly higher than those in sensitive OVCAR5 cells, suggesting a significantly upregulated FAO in resistant cells (Fig. 5e). Further, Seahorse measurement of OCR in PEO1 and PEO4 cells supports a significant increase of FAO rate in PEO4 cells compared to PEO1 cells (Supplementary Fig. 6b). These data indicate that FAO significantly increases in cisplatin-resistant cells.

To test whether the increased FAO contributes to cisplatin resistance, we firstly tested the response to etomoxir in cisplatin-resistant cells and parental cells. We observed higher sensitivity to etomoxir treatment in cisplatin-resistant cell lines when compared to their parental cell lines, in paired cell lines, including PEO1 and PEO4 (Fig. 5f), OVCAR5 and OVCAR5-cisR (Fig. 5g), and COV362 and COV362-cisR

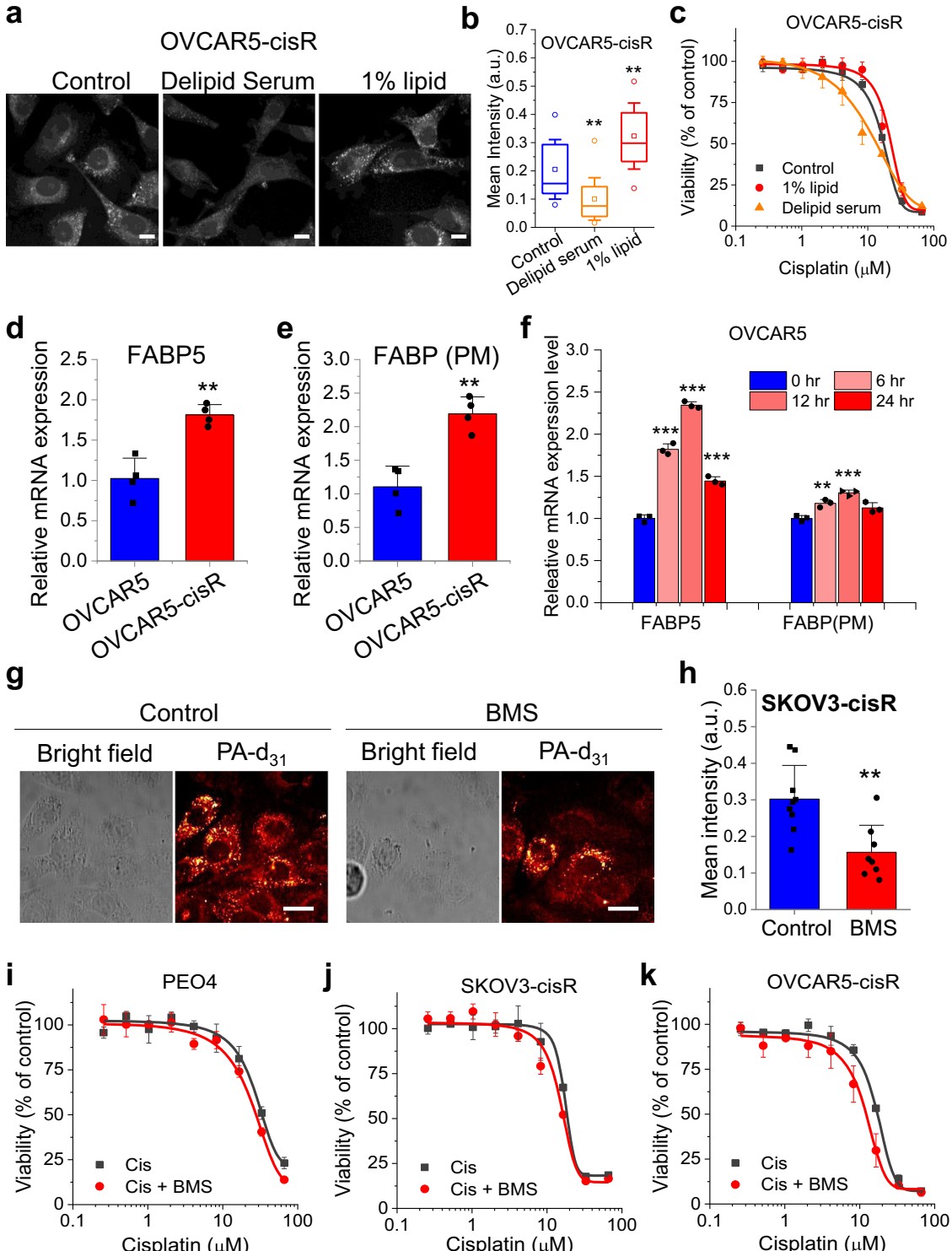

**Fig. 4 | FA uptake directly contributes to cisplatin resistance. a** SRS image of OVCAR5-cisR cell cultured with control serum (FBS), delipid serum or control serum supplemented with 1% lipid mixture for 24 h. **b** Quantitative C-H signal from lipid droplet for **a**. The outer box, inner box, lines, whiskers and circles indicates 25% to 75% of data, mean, medium, SD, maxima and minima respectively. $n = 15$ technical replicates. $P = 0.0043$ and 0.0072. **c** Dose-response to cisplatin under culture environment with control, reduced (medium containing delipid serum) and increased (control serum supplemented with 1% lipid mixture) lipid content for OVCAR5-cisR cells. **d, e** Relative mRNA expression level of FABP5 (d) and FABP(PM) (**e**) in OVCAR5 and -cisR cells. $n = 4$. $P = 0.0037$ and 0.0018. **f** Relative mRNA expression level of FABP5 and FABP(PM) in OVCAR5 cells treated with cisplatin for 0, 6, 12 or 24 h. $n = 3$. $P = 0.00016$, $2.4 \times 10^{-6}$,

0.00037, 0.0069, 0.00037 and 0.052. For the mRNA expression levels measurement (**d**–**f**), the results are shown as means + SD; n represents biological replicates. **g** Representative bright field and SRS images of SKOV3-cisR cell after BMS treatment at 10 μM for 24 h during concomitant incubation with 100 μM PA-$d_{31}$ for 6 h. **h** Quantification of C-D SRS signal intensity for (**g**). $n = 9$ and 8 technical replicates. $P = 0.0026$. **i**–**k** Dose-response to cisplatin with or without supplemental BMS treatment for PEO4 (**i**), SKOV3-cisR (**j**) and OVCAR5-cisR (**k**) cell. The results in all the does-response curves (**c** and **i**–**k**) are shown as means ± SD; $n = 3$ biological replicates. Data in all the bar charts (**d**–**f** and **h**) are shown as means + SD. All statistical significance was analyzed using two-sided Student's t test. **$P < 0.01$, and ***$P < 0.001$. All scale bar: 20 μm. Source data are provided in Source Data file.

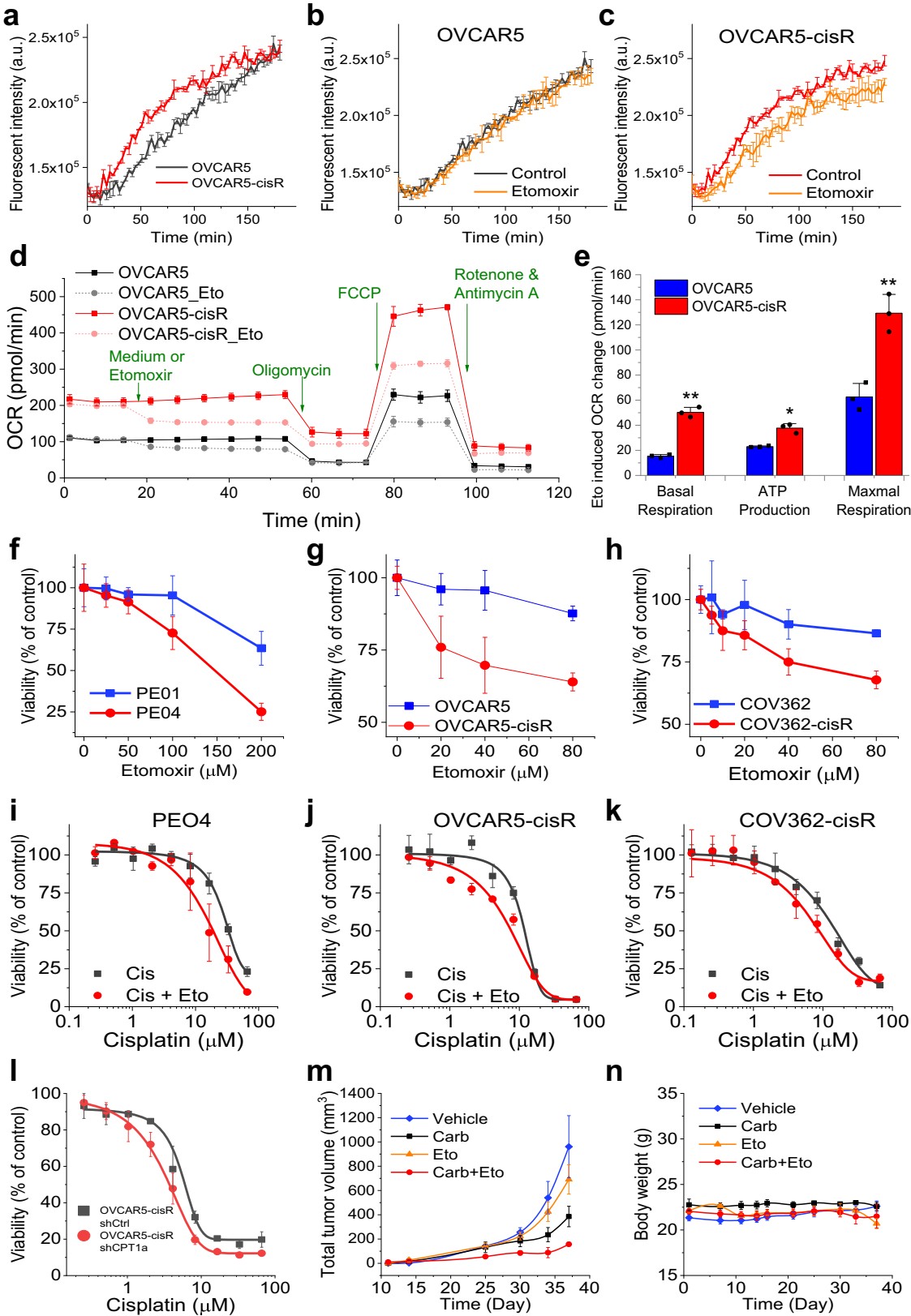

(Fig. 5h), indicating higher dependence on FAO in cisplatin-resistant cells. Next, we tested whether etomoxir treatment could reduce resistance to cisplatin. Our data show that the dose-response to cisplatin curves were significantly left shifted with etomoxir treatment in PEO4 (Fig. 5i), OVCA5-cisR (Fig. 5j), and COV362-cisR cells (Fig. 5k) and support potentially using etomoxir to re-sensitize resistant OC cells to

cisplatin. The observation was further confirmed by shRNA-mediated knockdown of *CPT1a* in OVCAR5-cisR cells (Supplementary Fig. 6c, d). Knockdown of *CPT1a* in OVCAR5-cisR cells increased its sensitivity to cisplatin treatment compared to the control group (Fig. 5l). Interestingly, the *CPT1a* mRNA and protein levels were similar in OVCAR5 and OVCAR5-cisR cells (Supplementary Fig. 6e, f), which implies that

**Fig. 5 | FA uptake contributes to cisplatin resistance by increasing FAO.**
**a** Oxygen consumption curves of OVCAR5-cisR and OVCAR5 over 3 h. $n = 4$ biolo-
gical replicates. **b**, **c** Oxygen consumption curves of OVCAR5-cisR (**b**) and OVCAR5
(**c**) with 40 μM etomoxir treatment over 3 h. $n = 4$ biological replicates. **d** Seahorse
measured OCR profile of OVCAR5 and OVCAR5-cisR cells with or without etomoxir
treatment, followed by injections of mitochondrial respiration inhibitors oligo-
mycin, FCCP, rotenone and antimycin A indicated by arrows. $n = 3$ technical repli-
cates. **e** Quantified etomoxir induced basal respiration, ATP production and
maximal respiration reduction in OVCAR5 and OVCAR5-cisR cell. Data are pre-
sented as means + SD; $n = 3$ technical replicates; two-sided Student's $t$ test;
$P = 0.0021, 0.018$ and $0.0046$; *$P < 0.05$ and **$P < 0.01$. **f**–**h** Dose-response to eto-
moxir for cisplatin resistant cell lines and their parental cell lines including PEO1

and PEO4 (**f**), OVCAR5 and -cisR (**g**) and COV362 and -cisR (**h**). **i**–**k** Dose-response to
cisplatin with or without supplemental etomoxir treatment at 40 μM for PEO4 (**i**),
OVCAR5-cisR (**j**) and COV362 -cisR (**k**) cell. **l** Dose-response to cisplatin for OVCAR5-
cisR shCtrl and shCPT1a cell. $n = 3$ biological replicates for does-response curves
(**f**–**l**). The data in all curves chart (**a**–**d** and **f**–**l**) are shown as means ± SD. **m** Total
tumor volume growth curve from day 14 to 37 after tumor cell inoculation for
vehicle ($n = 3$), carboplatin ($n = 3$), etomoxir ($n = 4$) and combinational ($n = 6$)
treatment groups. **n** Mice body weight record since tumor inoculation for vehicle
($n = 3$), carboplatin ($n = 3$), etomoxir ($n = 4$) and combinational ($n = 6$) treatment
groups. The data for PDX in vivo experiment (**m** and **n**) are shown as means ± SEM;
$n$ represents the number of animals. Source data are provided in Source Data file.

enhanced FAO in cisplatin-resistant OC is likely the result of increased
activation (not upregulation) of CPT1a.

To determine whether the functional changes are caused by
transcriptional reprogramming, RNA sequencing compared resistant
and parental OVCAR5 and SKOV3 cells. Heatmaps show hierarchical
clustering of genes related to FA metabolism and illustrate distinct
separation between sensitive/resistant cells (Supplementary Fig. 7a, b).
In both cell lines, several FAO-related genes, including *CRAT, PPARA,
ACOT8, HSD17B10, ACADVL, ACOX1* and *DECR1* were upregulated in
resistant cells, while FA synthesis-related genes such as *ME1, NSDHL,
DHCR24, FASN, ELOVL5, ALDH3A2, ACSL4* and *SERINC1* were down-
regulated (Supplementary Fig. 7a, b and Supplementary Data 1). These
transcriptomic findings support the observations indicating increased
FAO activity and reduced de novo FA synthesis in cisplatin-resistant OC.

To test whether interruption of FAO could sensitize ovarian
tumors to platinum in vivo, we used a patient-derived xenograft (PDX)
model rendered platinum resistant through repeated exposure to
carboplatin in vivo. To avoid toxicity induced by cisplatin, we sub-
stituted cisplatin with carboplatin, a second-generation agent. Carbo-
platin or etomoxir single-agent treatment induced a slight reduction of
tumor growth, whereas the combination treatment caused a sig-
nificant suppression of tumor growth (Fig. 5m). Body weights
remained stable in all groups, suggesting that the combination treat-
ment was tolerable (Fig. 5n). These data collectively support the
development of a combination of platinum with a FAO inhibitor for
platinum-resistant cancer treatment.

## FAO facilitates cancer cell survival under cisplatin-induced oxidative stress

Cisplatin has been known to cause cytotoxicity by inducing oxidative
stress, in addition to DNA adduct formation[47–49]. Excess oxidative
stress can inhibit glycolysis by inactivating key glycolytic enzymes,
such as pyruvate kinase M2 (PKM2) and glyceraldehyde 3-phosphate
dehydrogenase (GAPDH)[15,50]. Increased reactive oxidative species
(ROS) oxidizes intracellular NADPH and thus suppresses de novo
lipogenesis, as NADPH is one of the precursors for lipogenesis[51].
Therefore, we hypothesize that FA uptake and oxidation promote
cancer cell survival under cisplatin-induced oxidative stress by
replenishing free FA and ATP, deficiency of which is caused by
decreased de novo lipogenesis and glycolysis under oxidative stress.
To test this hypothesis, we first checked the oxidative stress level by
measuring intracellular ROS by using a fluorescent probe, 2′,7′-
dichlorodihydrofluorescein diacetate (DCFDA). By confocal micro-
scopy, we found that OVCAR5-cisR cells showed much stronger
fluorescent signal than OVCAR5 cells (Fig. 6a, b). Similar trend of
increased ROS in PEO4 cells compared to PEO1 cells was also observed
(Supplementary Fig. 8a, b). Furthermore, we analyzed the change in
ROS in OVCAR5 and OVCAR5-cisR cells treated with cisplatin and
found that cisplatin treatment induced significant increase of ROS
production in both cell lines (Fig. 6c).

Next, we examined whether the reduced form of intracellular
NADPH is depleted in cisplatin-resistant cells. NADPH/NADP$^+$ ratios

were significantly lower in cisplatin-resistant PEO4 (Fig. 6d) and
OVCAR5-cisR cells (Fig. 6e), when compared to PEO1 and OVCAR5
cells, respectively. The reduced NADPH level corroborated our SRS
images showing decreased de novo lipogenesis in cisplatin-resistant
cells. We then analyzed changes in glycolysis by measuring extra-
cellular acidification rate (ECAR) by Seahorse. As anticipated, cisplatin
treatment promptly lowered the ECAR rate in both PEO1 and PEO4
cells (Supplementary Fig. 8c), reaching significant reduction within
~30 min of treatment (Supplementary Fig. 8d). On the contrary, cis-
platin treatment also induced slight increase of OCR in PEO1 and PEO4
cells (Supplementary Fig. 8e, f). In agreement with the observed
decreased glycolysis, glucose uptake, measured by 2-NBDG under a
confocal microscope, was reduced in cisplatin-treated OVCAR5 cells
(Fig. 6f, g). Further, OVCAR5-cisR cells took up much less 2-NBDG than
OVCAR5 cells (Fig. 6f, g), implying a decreased reliance on glucose
metabolism in cisplatin-resistant OC cells.

With glycolysis suppressed by increased oxidative stress, we
suspected that ATP production would be impaired in cisplatin-
resistant cells or cisplatin-treated cells. We measured the cellular
ATP/ADP level and found the ratio of ATP/ADP was significantly lower
in both PEO4 (Fig. 6h) and OVCAR5-cisR cells (Fig. 6i), when compared
to PEO1 and OVCAR5 cells, respectively. Furthermore, acute treatment
with cisplatin reduced ATP/ADP ratio in OVCAR5 cells, but not in
OVCAR5-cisR cells. Supplementation with palmitic acid significantly
increased the ATP level in OVCAR5-cisR cells, but not in OVCAR5 cells
(Fig. 6j). Collectively, these data suggest that cisplatin-resistant OC
cells undergo metabolic reprogramming from glucose-dependent to
FA-dependent metabolism. This could be related to the propensity of
ovarian tumors to grow and disseminate in an adipocyte-rich micro-
environment. Adipocytes have been reported to provide fatty acids as
energy source for OC cells[12] and undergo increased lipolysis in
response to cisplatin treatment[52]. In summary (Fig. 6k), glycolysis and
lipogenesis are inhibited by cisplatin-induced oxidative stress, limiting
the production of energy, as well as the synthesis of free FAs. To sur-
vive and proliferate under cisplatin-induced oxidative stress, cancer
cells upregulate FA uptake and oxidation as an alternative route of
energy production.

## Cisplatin treatment induces a transient metabolic shift toward increased FA uptake in multiple types of cancers

Understanding that increased FA uptake in cisplatin-resistant OC cells
is likely a stable metabolic adaptation to cisplatin-induced oxidative
stress, we next tested if the same metabolic shift occurs in other types
of cancers upon cisplatin treatment. Platinum is widely used across
malignancies. Therefore, we selected a few representative cancer cell
lines, including MIA PaCa-2 pancreatic cancer, A549 lung cancer and
MD-MBA231 breast cancer, to test whether acute cisplatin treatment
changes the rate of FA uptake. We determined the IC$_{50}$ to cisplatin in
these three cell lines and selected 6.6 μM as the final treatment con-
centration, at which dose no significant cell death was induced (Sup-
plementary Fig. 9a-c). Results show that treatment with 6.6 μM
cisplatin significantly increased uptake of PA-d$_{31}$ and OA-d$_{34}$ in MIA

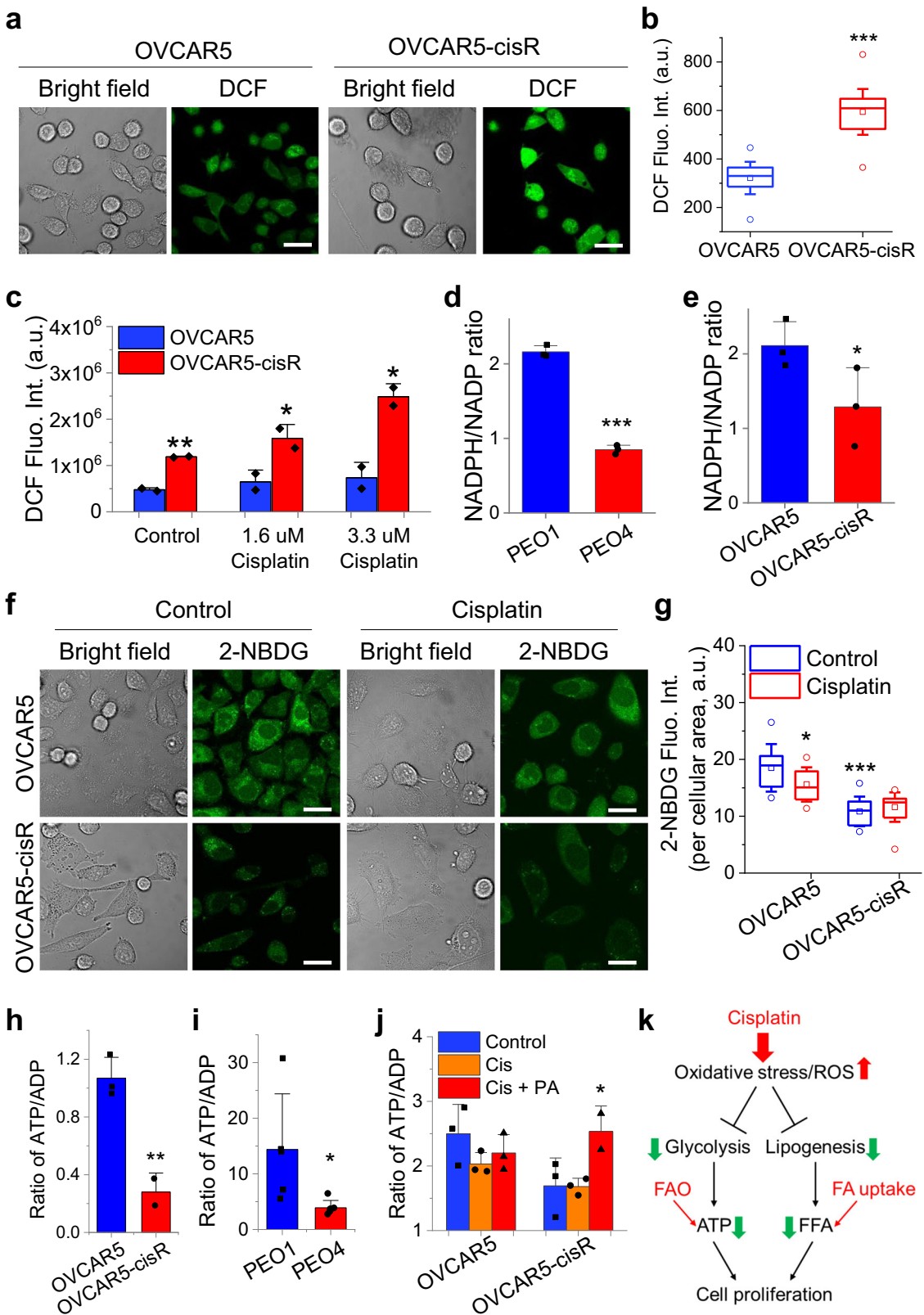

PaCa-2 cells (Fig. 7a, b). It is also worth noting that the fold increase in PA uptake was more significant than OA (Supplementary Fig. 9d), suggesting PA might be a preferred source of FA for cells under cisplatin-induced oxidative stress. Similarly, we observed that treatment with cisplatin also induced a significant increase in PA-d$_{31}$ and OA-d$_{34}$ uptake in A549 (Fig. 7c, d and Supplementary Fig. 9e) and MD-MBA231 cells (Fig. 7e, f and Supplementary Fig. 9f). These data suggest that our findings are broadly applicable to multiple types of cisplatin-resistant cancers.

## Discussion

Cisplatin and other platinum-based drugs, such as carboplatin and oxaliplatin, are widely used chemotherapy agents for multiple types of cancers, including ovarian, testicular, bladder, head and neck,

**Fig. 6 | Increased FA uptake and oxidation supports cancer cell survival under cisplatin-induced oxidative stress. a** Representative bright field and fluorescent images of OVCAR5 ($n = 55$) and OVCAR5-cisR ($n = 49$) treated with DCFDA cellular ROS assay kit. **b** Quantification of fluorescent signal intensity for (**a**). $P = 4.6 \times 10^{-29}$. **c** Quantification of DCF fluorescent signal intensity of OVCAR5 and -cisR with cisplatin treatment at 1.6 μM or 3.3 μM for 24 h. $n = 2$. P = 0.0064, 0.040 and 0.016. **d, e** Quantified NADPH/NADP ratio of PEO1 and PEO4 (**d**), and OVCAR5 and -cisR (**e**). $n = 3$. $P = 2.4 \times 10^{-5}$ and 0.048. **f** Representative bright field and fluorescent images of OVCAR5 and -cisR with 100 μM glucose analog 2-NBDG treatment for 2 h after incubation with 3.3 μM cisplatin for 24 h. **g** Quantified fluorescent signal intensity for (**f**). $n = 13, 16, 16$ and 17. $P = 0.019$ and $8.2 \times 10^{-7}$. **h, i** Quantified ATP/ADP ratio of cisplatin-resistant cell lines and their parental cell lines involving OVCAR5 ($n = 3$)

and -cisR ($n = 2$) (**h**), and PEO1 ($n = 5$) and PEO4 ($n = 6$) (**i**). $P = 0.0071$ and 0.039. **j** Quantified ATP/ADP ratio of OVCAR5 and -cisR treated with 3.3 μM cisplatin with or without supplement of 100 μM palmitic acid for 6 h. $n = 3$. $P = 0.046$ (**k**) Proposed mechanism about cisplatin effect on cellular metabolism and cell proliferation. All $n$ in fluorescent measurement (**a–c** and **f–g**) represents technical replicates, $n$ in the assay measurement (**d, e, h–j**) represents biological replicates. All Scale bar: 30 μm. For box plots (**b** and **g**), the bound of outer box indicates 25% to 75% of data; inner box indicates mean; lines represent medium; whiskers indicate SD; circles indicate maxima and minima of data. Data in all the bar charts (**c–e** and **h–j**) are shown as means + SD. Statistical significance was analyzed using one-sided Student's $t$ test. *$P < 0.05$, **$P < 0.01$, and ***$P < 0.001$. Source data are provided in Source Data file.

nonsmall-cell lung cancer and others[27]. Despite the high response rate following initial treatment, the effects of cisplatin and carboplatin are limited by severe side effects and high probability of drug resistance development[27,53]. In view of the high prevalence of cisplatin or carboplatin use in cancer management, there is a critical need to develop effective therapeutic options to overcome the resistance.

The major mechanism-of-action of cisplatin is formation of DNA adducts, which block transcription and DNA synthesis, while at the same time, activate DNA damage response mechanisms and mitochondrial detoxification mechanisms. Apoptosis eventually ensues if DNA lesions are not repaired, and oxidative stress is not buffered. Numerous efforts have been devoted to elucidating the mechanisms of cancer cell resistance to cisplatin. Most of these studies focused on adduct formation and subsequent activation of cell death pathways, for example, reduced formation of DNA adducts due to altered uptake/efflux, enhanced DNA damage repair, or impaired mitochondrial apoptosis pathway after adduct formation[53]. Other mechanisms of cisplatin resistance have received much less attention. Several studies have shown that cisplatin can have another mechanism-of-action by inducing oxidative stress in ovarian[47,54], prostate[55], and lung cancer[56]. Yet, little is known regarding how the cancer cells' defense against oxidative stress contributes to cisplatin resistance.

A few recent studies highlighted an association between metabolic reprogramming and cisplatin resistance. Alterations in glycolysis pathway were associated with cisplatin-generated oxidative stress in head and neck squamous cell carcinoma[57]. Lipid droplet production mediated by lysophosphatidylcholine acyltransferase 2 is linked to resistance to oxaliplatin in colorectal cancer[24]. More recently, adipocyte-induced *FABP4* upregulation was found to mediate carboplatin resistance in OC[44]. In this study, by using a high-throughput hyperspectral SRS imaging platform, we identified a metabolic switch from glucose to FA-dependent anabolic and energy metabolism in cisplatin-resistant cancer cells to adapt to cisplatin-induced oxidative stress. As depicted in Supplementary Fig. 10, our study reveals decreased glucose uptake, glycolysis and de novo lipogenesis while FA uptake and oxidation are increased, suggesting a central metabolic switch in anabolic and energetic metabolism in cisplatin-resistant OC cells. Inhibition of FAO re-sensitizes cisplatin-resistant OC cells to cisplatin treatment both in vitro and in vivo, paving the foundation toward a combinational therapy of FAO inhibitors and platinum drugs. We further show that cisplatin treatment induces similar metabolic shift in multiple other types of cancer, implying broad applicability of this strategy.

FAO, as an alternative path for energy production, has been shown to be upregulated in certain conditions, such as under metabolic stress[58]. However, it is less known whether and how FAO is deregulated in drug-resistant cancer cells. Our data support that cisplatin-induced oxidative stress is the culprit of increased FA uptake and oxidation. Oxidative stress depletes intracellular antioxidants, such as NADPH, and thus suppresses de novo lipogenesis, which requires NADPH. Decreased de novo lipogenesis could result into a lower level of malonyl-CoA, which is an allosteric inhibitor of CPT1, and

may trigger higher activity of CPT1[59,60]. This could be at least partially responsible for the observed increased FAO activity. However, as our RNA-seq analysis revealed, transcriptional regulation of other fatty acid metabolism genes could be also involved, requiring further investigation. Besides FAO, the process of FA uptake could represent another target for overcoming cisplatin resistance. Notably, the regulation of FA uptake involves multiple and redundant transporters, binding proteins and carrier proteins[42,43,45,61], and FA uptake contributes to several mechanisms significant for tumor survival and growth, including membrane biogenesis, FA pool replenishment, and ER stress prevention[62,63]. Future work is needed to completely elucidate the specific target that mediates the increased FA uptake and other mechanisms that benefit from enhanced FA uptake in cisplatin-resistant cancer cells.

Aside from pointing towards a potential therapeutic strategy for cisplatin-resistant cancers, ex vivo quantitative metabolic imaging of de novo lipogenesis and FA uptake in tumor cells could represent a functional marker for cisplatin responsiveness in clinical specimens at the single cell level. The conventional methods to determine cisplatin resistance rely on cell viability assays or measurement of various protein markers, which are time-consuming and lack accuracy. The metabolic imaging approach reported here provides a fast, functional, and quantitative way to determine cisplatin resistance based on functional metabolic signatures in resistant cells. This concept was validated in primary human ovarian cancer cells, showing high sensitivity for distinguishing platinum-sensitive and resistant tumors. While further testing in a larger tumor set is needed, particularly to determine whether metabolic differences between acquired and inherent type of resistance exist, our results put forward a quantitative assay with direct clinical applicability.

## Methods
Our research complies with all relevant ethical regulations. Animal studies were approved by the Institutional Animal Care and Use Committee (IACUC) at Northwestern University and were performed in the Developmental Therapeutics Core (DTC) of the Lurie Cancer Center.

### Cell lines
SKOV3 (Cat#: HTB-77), MIA PaCa2 (Cat#: CRL-1420), MDA-MB-231 (Cat#: CRM-HTB-26) and A549 (Cat#: CCL-185) cells were purchased from the American Type Culture Collection (ATCC) and PEO1 (Cat#: 10032308) and PEO4 (Cat#: 10032309) were from Sigma Aldrich. OVCAR5 cells were a generous gift from Dr. Marcus Peter, Northwestern University, and COV362 cells were from Dr. Kenneth Nephew, Indiana University. All cell lines were authenticated and tested to be mycoplasma negative. Mia Paca2 and A549 are in the list of known misidentified cell lines maintained by the International Cell Line Authentication Committee, while their authentication was performed by ATCC through STR profiling. The resistant cell lines SKOV3-cisR, COV362-cisR and OVCAR5-cisR were generated by treatment with 3 or 4 repeated or increasing doses of cisplatin for 24 h. Surviving cells were

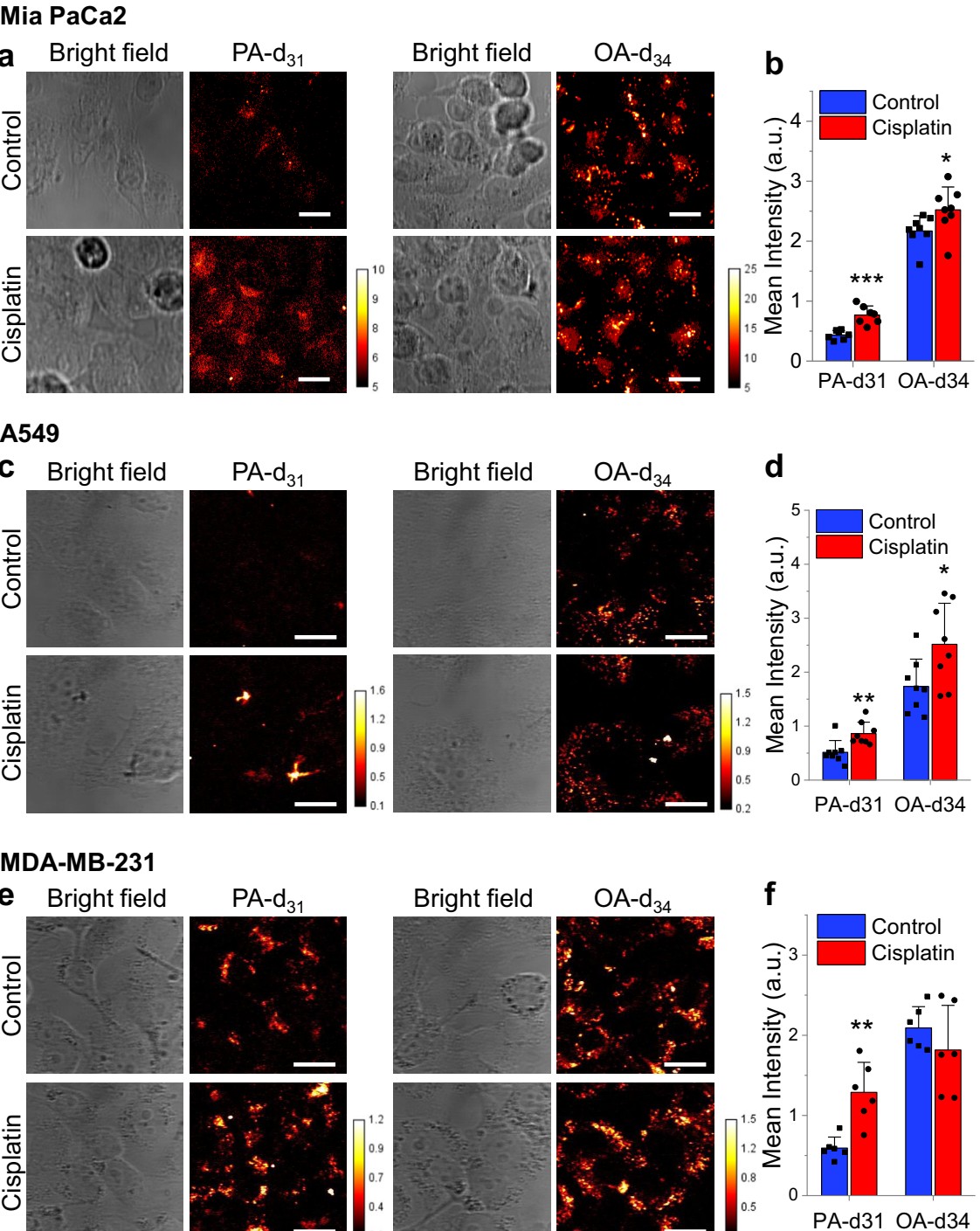

**Fig. 7 | Cisplatin induced FA uptake is a universal metabolic feature in multiple types of cancers. a** Representative bright field and SRS images of MIA PaCa2 cells treated with 6.6 µM cisplatin for 24 h followed by 100 µM PA-$d_{31}$ or OA-$d_{34}$ incubation for 6 h. **b** Quantitation of C-D signal in MIA PaCa-2 cells treated with or without cisplatin by mean intensity. $n = 7$ for PA-$d_{31}$ and $n = 8$ for OA-$d_{34}$. $P = 0.00029$ and 0.026. **c** Representative bright field and SRS images of A549 cells treated with 13.2 µM cisplatin for 48 h followed by 100 µM PA-$d_{31}$ or OA-$d_{34}$ incubation for 6 h. **d** Quantitation of C-D signal in A549 cells treated with or without cisplatin by mean intensity. $n = 8$. $P = 0.0031$ and 0.016. **e** Representative bright field and SRS images of MDA-MB-231 cells treated with 6.6 µM cisplatin for 24 h followed by 100 µM PA-$d_{31}$ or OA-$d_{34}$ incubation for 6 h. **f** Quantitation of C-D signal in MDA-MB-231 cells treated with or without cisplatin by mean intensity. $n = 6$. All n represents technical replicates. $P = 0.0024$. Data in all bar charts (**b, d, f**) are shown as means + SD; All statistical significance was analyzed using one-sided Student's t test. *$P < 0.05$. **$P < 0.01$. ***$P < 0.001$. Scale bar: 20 µm. Source data are provided in Source Data file.

allowed to recover for 3 to 4 weeks before receiving the next treatment. Changes in resistance to platinum were estimated by calculating half maximal inhibitory concentration (IC$_{50}$) values as described below[40]. PEO1, PEO4, OVCAR5, and OVCAR5-cisR cells were cultured in RPMI 1640 medium supplemented with 2 mM L-glutamine, 10% FBS and 100 units/mL penicillin/streptomycin. SKOV3, SKOV3-cisR, COV362, COV362-cisR and Mia Paca2 cells were cultured in high-glucose DMEM medium supplemented with 10% FBS and 100 units/mL penicillin/streptomycin. MDA-MB-231 and A549 cells were cultured in Leibovitz's L-15 medium and Kaighn's Modification of Ham's F-12

Medium, respectively supplemented with 10% FBS and 100 units/mL penicillin/streptomycin. For *CPT1a* knockdown cell lines development, cells were transfected with *CPT1a* or control shRNA lentiviral particles (Sigma Aldrich, TRCN0000036282) for 48 h and selected by 1 μg/ml puromycin for one week. All cells were cultured at 37 °C in a humidified incubator with 5% $CO_2$ supply.

## Materials

Glucose-d7, palmitic acid-d31 (PA-d31) and oleic acid-d34 (OA-d34) were purchased from Cambridge Isotope Laboratory. 17-Octadecynoic Acid (ODYA), BMS309403, cisplatin, and etomoxir were purchased from Cayman Chemicals. For treatment with cisplatin, 3.3 μM was used as the final concentration, unless otherwise specified.

## Primary human cells

De-identified high-grade serous ovarian tumors (HGSOC) and malignant ascites fluid specimens from consenting OC patients were obtained at the time of cytoreductive surgery either upfront or after neoadjuvant chemotherapy (interval debulking surgery) at the Northwestern University School of Medicine under an IRB approved protocol (STU00202468). All patients were followed prospectively and received platinum and taxane standard of care chemotherapy. Platinum resistance was defined as disease recurring within 6 months from completing carboplatin-based chemotherapy, as assessed clinically, by CA125 criteria or CT scans. Tumor tissues were enzymatically disassociated into single cell suspensions and cultured as previously described[64,65]. After centrifugation at 200 g for 5 min, 25,000 ascites derived tumor cells were cultured as monolayers in DMEM medium supplemented with 10% FBS and antibiotics prior to SRS imaging.

## In vivo experiments

The platinum-resistant PDX model was developed from a consenting patient with OC as previously described[66]. After passage through a donor animal, fresh tumor (equal size) was implanted subcutaneously (SC) in 20 female 7–8-week-old NSG mice (Jackson Labs, Cat#: JAX:00555). Tumor sizes were measured using calipers twice per week and tumor volumes were calculated according to the formula length x width$^2$/2. When the tumor volume reached 100 mm$^3$, the animals were randomized into four groups: vehicle, carboplatin alone (10 mg/kg weekly i.p. injection), etomoxir alone (40 mg/kg daily i.p. injection), and combination of carboplatin (10 mg/kg weekly i.p. injection) and etomoxir (40 mg/kg daily i.p. injection). Body weights and habitus were monitored twice per week.

Xenografts were obtained through intraperitoneally implantation of 2 million OVCAR5 cells in female (6–8 weeks old) athymic nude mice (Foxn1$^{nu}$, Envigo). Two weeks after inoculation, tumor harboring mice were treated with PBS (sensitive group, $n = 3$) or 25 mg/kg carboplatin (resistant group, $n = 3$) via weekly i.p. injection for three cycles. Tumors were collected and weighted one week after the last cycle and frozen. For SRS imaging, tumors were sectioned at 5–10 nm thickness slices by cryostat. To isolate tumor cells from tissue, PBS and carboplatin-treated xenografts were mechanically and enzymatically dissociated in Dulbecco's modified Eagle's medium/F12 (Thermo Fisher Scientific) containing collagenase (300 IU/ml, Sigma-Aldrich) and hyaluronidase (300 IU/ml, Sigma-Aldrich) for 2–4 h at 37 °C. Red blood cell lysis used RBC lysis buffer (BioLegend), followed by DNase (Qiagen) treatment and filtering through a 40 μm cell strainer (Fisher Scientific) to yield single cells suspension, which were examined for responsiveness to cisplatin ex vivo.

For all animal experiment, mice were housed at 21 °C –23 °C with a 12/12 dark/light cycle. The humidity of the housing environment is 35%. Mice were sacrificed when the largest tumor exceeded 1500 mm$^3$ or if human endpoints were reached earlier. The mouse diet Cat# is 7912 from Teklad/Envigo. The chow is always available (ad libitum) to the mice for consumption.

## Large-area hyperspectral stimulated Raman scattering (SRS) imaging

Hyperspectral SRS imaging was performed on a lab-built system following previously published method[8]. The laser source is a femtosecond laser (InSight DeepSee, Spectra-Physics, Santa Clara, CA, USA) operating at 80 MHz with two synchronized output beams, a tunable pump beam ranging from 680 nm to 1300 nm and a Stokes beam fixed at 1040 nm. For imaging at the C-H vibration region (2800 – 3050 cm$^{-1}$), pump beam was tuned to 798 nm. The Stokes beam was modulated at 2.3 MHz by an acousto-optic modulator (1205-C, Isomet). After combination, both beams were chirped by two 12.7 cm long SF57 glass rods and then sent to a laser-scanning microscope. The power of pump and Stokes beam before the microscope was controlled to be 20 mW and 200 mW, respectively. A 60x water immersion objective (NA = 1.2, UPlanApo/IR, Olympus) was used to focus the light on the sample, and an oil condenser (NA = 1.4, U-AAC, Olympus) was used to collect the signal. For hyperspectral SRS imaging, a 50-image stack was acquired at different pump-Stokes temporal delay, which was controlled by tuning the optical path difference between pump and Stokes beam through a translation delay stage. Raman shift was calibrated using standard samples, including DMSO, oleic acid, and linolenic acid.

To achieve large-area mapping, samples were fixed on a motorized stage (PH117, Prior Scientific). A lab-built LabView-based program was used to control the moving of the stage and stitching of images. The stage moves to the adjacent location with partial overlap after a hyperspectral SRS image was acquired at current location. A montage image composed of 5 × 5 individual 400 × 400-pixel images was acquired at each area of interest. The size of the montage image is approximately 500 × 500 μm. The pixel dwell time is set as 10 μs. For each sample, at least 3 montage images were acquired at different area of interest.

## Spectral phasor and CellProfiler-based single cell analysis

The acquired large-area hyperspectral SRS images were segmented through Spectral phasor analysis modified from previously published method[39]. Spectral phasor was installed as a plugin in ImageJ. The images were transformed into a two-dimension phasor plot based on Fourier Transform. Each dot on the phasor plot represents an SRS spectrum at a particular pixel. Pixels with similar spectra or chemical content were clustered on the phasor plot. Nuclei and lipid images were generated by mapping the corresponding clusters on the phasor plot back to two separate images.

Lipid analysis in single cells were performed through the software CellProfiler[67]. The map of nuclei and cell images were input into CellProfiler to outline each individual cells. The lipid map was input into CellProfiler to pick up the lipid droplet (LD) particles. Then, the lipid map was masked onto the outlined cell map to label the lipids. Morphological information of each cell and lipid analysis, including LD number and intensity in single cells were measured and reported in the output results. The total lipid intensity in each cell was plotted as histogram graphs. For each sample, a few hundreds to a thousand of cells were analyzed.

## Isotope labeling and SRS imaging

For labeling with glucose-d7, media was replaced with glucose-free DMEM medium (Thermo Fisher Scientific, # 11966025) + 10% FBS + P/S supplemented with 25 mM glucose-d7 after seeding the cells in 35 mm glass-bottom dishes overnight. For labeling with FA or analogs, including PA-d31, OA-d34 and ODYA, FA or analogs were added to the culture media at final concentration of 100 μM and cells were treated for 6 h. For quantitative SRS imaging, cells on glass-bottom dishes were fixed with 10% neutral buffered formalin for 30 min and washed with PBS for 3 times. Hyperspectral SRS imaging was performed to the cells at Raman spectral region from 2100 to 2300 cm$^{-1}$.

## Reactive Oxidative Species (ROS) measurement

Cellular ROS was measured using a fluorescent probe, 2′,7′-Dichloro-fluorescin diacetate (DCFDA) (Sigma Aldrich). Cells seeded in glass-bottom dishes were treated with or without 3.3 μM cisplatin for 3 h. DCFDA was added to the medium at final concentration of 10 μM and incubated for 15 min. After washing with PBS for 3 times, cells were immediately imaged under confocal microscope (Zeiss LSM 700 microscope) with 488 nm as the exciting source. Laser power was controlled at low setting to avoid fluorescence extinction. Images at -10 field of view were acquired for each sample.

## FAO assay

FAO was measured using a commercial kit (Abcam, #ab217602) following the provided protocol. Briefly, cells were seeded in 96-well plate with 150k cells/well. After incubation overnight, medium was replaced. 10 μL of Extracellular $O_2$ consumption reagent, and 2 drops of high-sensitivity mineral oil (preheated at 37 °C) were added to each well. Fluorescence was measured in plate reader at 2 min intervals for 180 min at excitation/emission = 380/650 nm. Etomoxir was added at final concentration of 40 μM to block FAO. Oxygen consumption rate (OCR) is presented as $\Delta_{\text{Fluorescence intensity}}$/min/cell and FAO rate is calculated as $OCR_{FAO} = OCR_{total} - OCR_{Etomoxir}$. At least three replicates were included for each measurement.

## NADPH and ATP assay

NADP/NADPH and ADP/ATP were measured by using commercial kits (Abcam, #ab65349 and #ab65313). For NADPH measurement, cells (-1 × 10$^6$ cells) were pelleted and extracted using NADPH/NADP extraction buffer. Total NADP/NADPH was directly measured using the assay kit and NADPH alone was measured after decomposing NADP by heating at 60 °C for 30 min. Absorption at 450 nm was measured by plate reader (Molecular Devices, SpectraMax i3x). For ATP measurement, cells were seeded in 96-well plates. ATP was measured directly and total ATP + ADP was measured by converting ADP to ATP. The luminescent signal was measured by plate reader. At least three replicates were included for each measurement.

## Cell viability assay

Cell viability was measured by MTS assay (Abcam, #ab197010) or by CellTiter-Glo assay (Promega, #G7570). Cells were seeded at 96-well plates at densities of 2000-5000 cells per well overnight. Treatment was added to the cells at indicated concentrations for 72 h. Cell viability was measured by incubating with MTS reagent for 4 h and reading absorbance at 490 nm or incubating with CellTier-Glo reagent for 10 min and reading luminescence by a plate reader. Six replicates were used for each group.

## Glucose uptake assay

Glucose uptake was measured using a fluorescent glucose analog, (2-NBDG) (Cayman chemicals). Cells seeded in glass-bottom dishes were incubated with 100 μM 2-NBDG for 2 h. Fluorescent images were taken by confocal microscope (Zeiss LSM 700 microscope) with 488 nm laser as excitation source. Images at -10 field of view were acquired for each sample.

## Measurement of OCR and ECAR by Seahorse

Cell lines were seeded in a Seahorse XF96 Cell Culture Microplate (Agilent) at density of 6 × 10$^4$ (OVCAR5 pair) or 4 × 10$^4$ (PEO pair) per well. After incubation at 37 °C overnight for cell attachment, oxygen concentration rate (OCR) and extracellular acidification rate (ECAR) were measured through Seahorse XFe96 Analyzer (Agilent). Measurement time was 30 seconds following 3 minutes mixture and 30 seconds waiting time. First three cycles were used for basal respiration measurement. Effects of 4 μM oligomycin, 4 μM FCCP, 25 μM rotenone, 50 μM antimycin A, 26.4 μM cisplatin, or 40 μM etomoxir on OC cells OCR and ECAR were measured. Basal respiration, ATP production and maximal respiration reduction were calculated following the manufacturer's instructions.

## Reverse transcription-PCR (RT-PCR)

Total RNA from OC cell lines were extracted via RNeasy Mini Kit (Qiagen Inc.) and reverse transcribed by iScript cDNA Synthesis Kit (Bio-Rad). RT-PCR was performed through StepOne Plus RT-PCR (Applied Biosystems) using Power SYBR Green Master Mix (Thermo Fisher Scientific) All procedure was following manufactures' instructions. RT-PCR reaction generated a melting curve and cycle threshold (Ct) was recorded for the gene of interested and house-keeping control gene (PPIA). The relative RNA expression level was calculated as ΔCt and normalized by subtracting the Ct value of target gene from that of control gene. Results are presented as means + SD. Measurements were performed in biological triplicate and each biological replicate includes three technical replicates. Primer sequences are listed in Supplementary Table 3.

## Western blot

Proteins were extracted from cell culture by RIPA lysis buffer (Sigma Aldrich) with protease and phosphatase inhibitor cocktail and sample reducing agent (Thermo Fisher Scientific). Proteins were separated in Bolt Bis-Tris Plus gels (Thermo Fisher Scientific) through gel electro-phoresis and transferred to PVDF membrane (Bio-Rad). After blocking in 5% non-fat milk (Bio-Rad) for 1 h at room temperature, membranes were incubated with primary antibodies (CPT1a (1:1000) (Proteintech; Cat#: 15184-1-AP; RRID: AB_2084676) and GapDH (1:2000) (Proteintech; Cat#: 60004-1-Ig;RRID: AB_2107436; Clone#: 1E6D9) overnight at 4 °C followed by secondary anti-mouse antibodies (1:10000) (Proteintech; Cat#: SA00001-1;RRID: AB_2722565) for 1 h at room temperature. Protein bands were developed by ECL reagent (Thermo Fisher Scientific) and detected through ChemiDoc MP imaging system (Bio-Rad). The band intensity was determined using ImageJ. Full scan blots are in the Source Data file.

## RNA sequencing analysis

RNA-seq data from OVCAR-5 and SKOV-3 cisplatin-resistant vs parental cells were downloaded from the Gene Expression Omnibus with accession ID: GSE148003[40]. Raw data were normalized with the R package edgeR[68]. Overlapping genes in the Hallmark Fatty Acid Metabolism gene set between OVCAR5-cisR versus parental cells and SKOV3-cisR versus parental cells were used for generating heatmaps using the R package heatmap. Specifically, the heatmap of hierarchical clustering was generated for OVCAR5-cisR versus parental cells by using normalized counts. The same gene order after hierarchical clustering was applied to produce a heatmap for SKOV3-cisR versus parental cells.

## Quantification and Statistical Analysis

All the data are presented as means ± SD unless otherwise specified. The statistical significance was analyzed using two-tailed Student's t test. All experiments were repeated at least 3 times. N is indicated sample size for each experiment. $P < 0.05$ was considered statistically different. Statistical parameters can be found in figure legends. Data was analyzed and qualified by ImageJ, MATLAB, and Microsoft Excel. Origin was used for figure generation.

## Reporting summary

Further information on research design is available in the Nature Research Reporting Summary linked to this article.

## Data availability

The RNA-seq data used in this paper are available in the Gene Expression Omnibus with accession ID: GSE148003. Because the

amount and size of SRS imaging raw data are enormous, SRS imaging raw data related to this work is available upon request to the corresponding author (jxcheng@bu.edu (J.X.C.)) through cloud drive share. All the other data are available within the article and its Supplementary Information. Source data are provided with this paper.

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

## Acknowledgements

This work was supported by R01CA224275 to DM and JXC and R33CA223581 and R35GM136223 to JXC. Research reported in this publication was supported by the Boston University Micro and Nano Imaging Facility and the Office of the Director, National Institutes of Health of the National Institutes of Health under award Number S10OD024993. PDX experiments were performed in the Northwestern University –Center for Developmental Therapeutics supported by the Cancer Center Support Grant NCI CA060553.

## Author contributions

J.L., J.X.C. and D.M. co-designed the experiments. J.L., Y.T., H.C., G.Z., Y.W. performed the experiments. K.C.H. provided help in data analysis. J.L. and Y.T. co-wrote the manuscript. All authors read and edited the manuscript.

## Competing interests

The authors declare no competing interests.
