## [Peer Review File · Nature Communications]

Title: Metabolic Reprogramming from Glycolysis to Fatty Acid Uptake and beta-Oxidation in Platinum-Resistant Cancer CellsREVIEWER COMMENTS

Reviewer #1 (Remarks to the Author):

Utilizing high-throughput hSRS, deuterated probe incubation and spectral phasor segmentation, Li et al. nicely identified a metabolic switch from glycolysis to fatty acid uptake in four pairs of platinum sensitive and resistant ovarian cancer cells. They defined a “metabolic index” for quantification based on the C-D signal ratio from fatty acid uptake versus glucose incorporation and successfully extrapolated a linear correlation to cisplatin resistance. They also showed that this observation could be likely extended to lung, breast, and pancreatic cancer cells with acute cisplatin treatment. They further evaluated the fatty acid oxidation rate and drew a relationship between cisplatin-induced oxidative stress to elevated fatty acids uptake in cisplatin-resistant cancer cells. This is a very comprehensive report. While the demonstrations of the metabolic switch here are convincing and would potentially offer new therapeutic insights, there are a few conclusions that could not be fully supported by the data shown in the manuscript. The authors should address those concerns before the paper could be further considered for publication in Nat. Commun.

1, For the fatty acid uptake essay, the authors demonstrated 100 μ M PA-d31 and OA-D35 incubation for 6 h. How is this value determined? Is this concentration value close to the relative fatty acid concentrations in normal serum? Have the authors tried lower-concentration incubation? How does the author rule out the perturbation from this relatively high concentration of fatty acid incubation? Would it be possible that the cisplatin-resistance cells have worse intracellular regulation on fatty acid metabolism, which lead to the increased fatty acid signals?

Since the central discovery here is the metabolic switch, proper controls would be essential. For example, it has been shown that relatively high concentration incubation of PA may induce cellular stress and the formation of phase-separated domains in just a few hours (<https://www.pnas.org/content/114/51/13394>). I think it would at least be essential to show the data with lower concentration incubation here.

2. In Fig. 4g-i, the authors indicated that “Furthermore, inhibition of fatty acid uptake by BMS-309403 significantly reduced resistance to cisplatin in multiple resistant cell lines, including PEO4 (Fig. 4i), SKOV3-cisR (Fig. 262 4j), and OVCAR5-cisR cells (Fig. 4k), which supports a functional role of fatty acid uptake in the development of cisplatin resistance in ovarian cancer cells.” Quantitatively, how is “significantly” defined here? It is not clear to me how to draw the statistical significance from the data here. From the change of the IC50 value, the data does not seem to represent a significant shift. In addition, since the authors claim that the cisplatin-resistance cells shift toward FA uptake compared to that for the cisplatin-sensitive correspondents. Does this mean such “sensitization” effect in the cisplatin-resistance cells would be absent in the cisplatin sensitive cells? This control experiment is needed to make a relatively strong claim for the unique metabolic switch and to lay the foundation for the combinational therapy conclusion.

3. In Fig. 5, the authors showed the FAO inhibition results by etomoxir to show that a metabolic shift toward higher FAO in resistant cells. However, etomoxir has been suggested to show a potentially severe off-target effect when used beyond concentrations of 5-10 μM . Although up to 200 μM concentration has been shown in some reports, the caution for the evaluations of off-target influence has been discussed in other reports. The authors here used about 20-200 μM treatment for different cells. How was the off-target effect evaluated here? Although a shRNA experiment is shown in Fig. 5j, the curve seems to be rather noisy. A more robust statistical analysis should be done.

4. The authors claimed that they “developed a method” for the simultaneous ODA and glucose-d7 imaging. This is probably not a proper claim with the previous technical reports on these two tags for imaging.

(<https://pubs.rsc.org/en/content/articlelanding/2018/CC/C7CC08217G#!divAbstract>)

Reviewer #2 (Remarks to the Author):

In this manuscript, Junjie Li and colleagues use imaging and single cell analysis to analyze the metabolism of cisplatin-resistant cells in ovarian cancer cell lines and in primary cells. Their results suggest that increased fatty acid uptake allow cancer cells to survive cisplatin-induced oxidative stress.

Altogether, the data look really good to me and the manuscript is clearly written.

However, there are two issues that the authors may wish to consider to improve the manuscript.

1/ Major:

The mechanism by which cisplatin resistant cells reprogram fatty acid uptake and metabolism is not fully elucidated. It could be a nice addition to investigate whether cisplatin induces significant rewiring of gene expression in ovarian cancer cells and thereby determine the observed changes in lipid metabolism.

2/ Minor:

The authors mostly discuss fatty acid oxidation, but enhanced uptake of fatty acids might also be required for tumor growth (contributing to membrane biogenesis). If relevant, I think this should also be discussed.

It could also be interesting to investigate/discuss whether reduced lipogenesis influences fatty acid beta-oxidation through malonyl-coa mediated inhibition of mitochondrial CPT?

Reviewer #3 (Remarks to the Author):

The manuscript titled “Metabolic Reprogramming from Glycolysis to Fatty Acid Uptake and beta-Oxidation in Platinum-Resistant Cancer Cells” by Junjie Li and colleagues. The authors describe the mechanism of metabolic programming in platinum-resistant ovarian cancer cells. They report that cisplatin-resistant cells exhibit increased uptake of exogenous fatty acids, decreased glucose uptake and de novo lipogenesis, indicating a metabolic switch from glycolysis-dependent to fatty acid uptake/ beta-oxidation dependent anabolic and energy metabolism. The authors used high throughput stimulated Raman scattering imaging and single cell analysis metabolic to show the changes of the uptake of glucose and fatty acids in paired sensitive, and resistant ovarian cancer cells and developed a metabolic index incorporating measurements of glucose derived anabolism and fatty acid uptake and found that this index linearly correlates with cisplatin-resistance in established ovarian cancer cell lines and in primary cells isolated from ovarian cancer patients. They also performed mechanistic studies to prove that the increased fatty acid uptake facilitates cancer cell survival under cisplatin-induced oxidative stress by enhancing energy production through beta-oxidation.

The hypothesis is clear, and the experimental approach used to test the hypothesis is logic and clear.

Results are clearly presented with supporting supplementary data.

Overall the manuscript is well written but the material and method section needs extensive editing and clarification.

Reviewer #4 (Remarks to the Author):

COMMENTS TO THE AUTHORS

The manuscript by Lin and Tan et al describe the dependency and role of exogenous FA in ovarian cancer cells which can be exploited for detection of cisplatin resistance and design new treatment strategies. Detection and treatment of chemoresistance in OC is a huge hurdle and the current research is very relevant. The potential of using SRS imaging to detect chemoresistance at single cell level is exciting. The concept demonstrated by the authors is excellent however, the enthusiasm is greatly lowered due to lack of in vivo data and more vigorous experiments to demonstrate the acute dependency of chemoresistance to exogenous FAs, as many reports show chemo resistant cells to develop a flexible metabolism allowing them to take advantage of various fuel substrates under stress.

1. While the authors indicate that they are studying development of therapeutic chemoresistance, it is unclear if the problem addressed is being of inherent chemoresistance or acquired chemoresistance. Almost all the cell line models used are of artificially induced chemoresistance. The patient cells used have not been described well enough to know if these are from naïve patients or obtained from tumors that were exposed to chemoresistance. This may reveal if the FA dependency is adaptive metabolic alteration to development of resistance. In addition, for several years' carboplatin use has become the norm instead of cisplatin.

2. It will be more relevant to create resistance in vivo by isolating cells after cisplatin treatment rather

than the artificial in vitro dose incremental system. To demonstrate the true potential of SRS imaging to detect FAO at single cell level, these tumors can be used to show the heterogeneity of chemoresistance within the same tumor (Fig 1 demonstrated in tumors growth in mice +/- cisplatin). A confounding thought is that since post-chemo/recurrent tumor surgeries are not common, obtaining cancer cells to detect chemoresistance may be a hurdle to apply SRS imaging in tumor cells, and may have more applicability in inherent chemo resistant tumors.

3. To demonstrate that the increased lipid content of the chemo resistant cells is independent of de novo lipogenesis, demonstration that inhibition of de novo lipogenesis still results in increased lipid content/droplets in the resistant cells should be shown. Fig. 2 should be supported by functional measurements of glycolysis, mitochondrial oxidation, and FAO (probably by Seahorse) when the cells are grown in presence of glucose or exogenous FAs. Is the uptake of glucose vs FA dependent on the expression of transporters? Does GLUT1 expression gets decreased in resistant cells and in response to cisplatin while the FA transporters increase?

4. The applicability of the “metabolic index” is not clear. The in vitro created systems provide a very clear-cut correlation which may not apply to a heterogenous tumor. If the index score =0.5, how will the status of resistance be inferred?

5. The patient data (Fig. 3) should be supported by functional data showing increased FAO occurring in sensitive vs resistance cells.

6. Limiting exogenous FAs resulted in decreased lipid accumulation. In these conditions, was sufficient glucose provided? Did the cells adapt by intaking more glucose? Will providing alternative fuels (glucose/glutamine etc.) make the cells overcome the FA deficiency? This is important to address as many chemo resistant cells, including ovarian have been reported to possess a flexible metabolism allowing them to utilize different energy pathways in times of stress to survive.

7. It is unclear why the expression of FA transporters was examined in only the OVCAR5 cell line set and not in all the various cell line sets. It would be more relevant to show the expression profiles in PE01/4 cell lines and in the patient derived cell lines used in Fig. 3. The use of different cell lines to showcase different data throughout the manuscript is bothersome.

8. In Fig. 5, it is unclear why Seahorse FAO assay was not used to demonstrate specificity of FA induced OCR and its inhibition by etomoxir, as the authors clearly have access to the instrument. In the same experiment, rescue experiments with increased exogenous FAs and glucose should be presented to demonstrate the specificity of FAO and sensitivity.

9. The PDX experiment is well performed but may benefit by including additional PDX or xenograft models and the lipid accumulation demonstrated during the development of resistance. Including the FA transporter inhibitor BMS-309403 may strengthen the role of exogenous FA uptake, as omentum is known to be a source exogenous FAs in ovarian cancer.

10. Can the authors address the role of nutrient status in the tumor microenvironment. Recent work suggests that limiting nutrient (for example glucose or glutamine) may force the cancer cell to start using alternative fuel sources. Since the authors propose the affinity of chemo resistant cells towards FA over glucose, this should be discussed in context of ovarian cancer.

Response Letter

Dear Dr. Lombardo,

We appreciate the positive evaluation of our work and the opportunity to submit a revised manuscript. The reviewers were uniformly positive and described the manuscript as “comprehensive report”, “demonstration of metabolic switch is convincing”, “data look really good to me and the manuscript is clearly written”, “current research is very relevant”, “concept demonstrated by the authors is excellent”.

A few issues were raised, which we have addressed with additional experimentation or clarifications. New experimental data are included in the revised manuscript (new figures: Fig. 1g-h, Fig. 4f, Fig. 5d-e, Supplementary Fig. 1h-i, Fig 2c-d, Fig. 3f-g, Fig. 4e-f, Fig. 4i-n, Fig. 5c-e, Fig. 6a-b; new tables: Supplemental Tables 2 and 4; revised figures: Fig. 3a-c, Fig. 3g, Fig. 5l and Supplementary Fig.1h-i), as detailed in our response letter below. Because of the amount of data included the manuscript, some of the new data are not included in the final manuscript, but are included in this letter for the reviewers' evaluation.

Reviewer #1 (Remarks to the Author):

1, For the fatty acid uptake essay, the authors demonstrated 100 μ M PA-d31 and OA-D35 incubation for 6 h. How is this value determined? Is this concentration value close to the relative fatty acid concentrations in normal serum? Have the authors tried lower-concentration incubation? How does the author rule out the perturbation from this relatively high concentration of fatty acid incubation? Would it be possible that the cisplatin-resistance cells have worse intracellular regulation on fatty acid metabolism, which lead to the increased fatty acid signals?

Since the central discovery here is the metabolic switch, proper controls would be essential. For example, it has been shown that relatively high concentration incubation of PA may induce cellular stress and the formation of phase-separated domains in just a few hours (<https://www.pnas.org/content/114/51/13394>). I think it would at least be essential to show the data with lower concentration incubation here.

Response: We appreciate the comments. The reviewer has raised a good point. Indeed, high concentration of free fatty acid could induce cytotoxicity, as demonstrated in the abovementioned paper and other papers¹⁻³. However, it has been shown that 100 μ M palmitic acid or oleic acid are tolerated in living cells^{4,5}. Therefore, we believe the increased fatty acid signal in cisplatin-resistant cells is due to altered fatty acid metabolism, but not due to differences in cell state, such as membrane lipid composition or fluidity. To further confirm this, we performed FA uptake imaging and quantitative

analysis in SKOV3 and SKOV3-cisR cells fed with 10, 20 and 50 μM PA-d31 or OA-d34 for 6 h. When fed with 50 μM PA-d31 or OA-d34, resistant cell SKOV3-cisR still showed increased fatty acid uptake compared with sensitive cell SKOV3 (revised **Supplementary Fig.4 k-n**). When the fatty acid concentration was reduced to 20 μM , the small difference of OA-d34 uptake became less obvious due to the reduced signal intensity, but the PA-d31 uptake difference between sensitive and resistant cell lines remain statistically significant. (revised **Supplementary Fig.4 k-n**). CD signal from 10 μM deuterated fatty acid incubation in sensitive cell line is undetectable and thus cannot be quantified (data not shown).

These new data were added to the revised manuscript (page 13, first paragraph).

2. In Fig. 4g-i, the authors indicated that “Furthermore, inhibition of fatty acid uptake by BMS-309403 significantly reduced resistance to cisplatin in multiple resistant cell lines, including PEO4 (Fig. 4i), SKOV3-cisR (Fig. 4j), and OVCAR5-cisR cells (Fig. 4k), which supports a functional role of fatty acid uptake in the development of cisplatin resistance in ovarian cancer cells.” Quantitatively, how is “significantly” defined here? It is not clear to me how to draw the statistical significance from the data here. From the change of the IC50 value, the data does not seem to represent a significant shift. In addition, since the authors claim that the cisplatin-resistance cells shift toward FA uptake compared to that for the cisplatin-sensitive correspondents. Does this mean such “sensitization” effect in the cisplatin-resistance cells would be absent in the cisplatin sensitive cells? This control experiment is needed to make a relatively strong claim for the unique metabolic switch and to lay the foundation for the combinational therapy conclusion.

Response: In response, we adjusted our statement to reflect that the observed differences, albeit statistically significant, are modest. We also included the control experiments the reviewer suggested. The data show that the sensitization effect of BMS is diminished at least in some sensitive cell lines, such as SKOV3 cells, when compared to resistant cell lines (revised **Supplementary Fig.5c-e**). Such a difference in PEO1 and OVCAR5 cells is not clear. It may reflect the heterogeneity between cell lines. We have added this new data to the revised manuscript (page 14, first paragraph).

3. In Fig. 5, the authors showed the FAO inhibition results by etomoxir to show that a metabolic shift toward higher FAO in resistant cells. However, etomoxir has been suggested to show a potentially severe off-target effect when used beyond concentrations of 5-10 μM . Although up to 200 μM concentration has been shown in some reports, the caution for the evaluations of off-target influence has been discussed in other reports. The authors here used about 20-200 μM treatment for different cells. How was the off-target effect evaluated here? Although a shRNA experiment is shown in Fig. 5j, the curve seems to be rather noisy. A more robust statistical analysis should be done.

Response: We agree with the comments. For a rigorous analysis, we repeated the cisplatin dose response experiment in control or CPT1a shRNA knockdown cells in

three replicates. Our new data confirmed a significant sensitization effect from CPT1a knock down. See revised **Fig. 5I**.

4. The authors claimed that they “developed a method” for the simultaneous ODYA and glucose-d7 imaging. This is probably not a proper claim with the previous technical reports on these two tags for imaging.

(<https://pubs.rsc.org/en/content/articlelanding/2018/CC/C7CC08217G#!divAbstract>)

Response: We agree that both glucose-d7 and alkyne are widely used in SRS imaging. Here, we adopted these probes for SRS imaging assessment of cisplatin resistance. Thus, we have revised the manuscript accordingly (page 10, second paragraph). Previous literatures are cited (page 4, third paragraph and page 10, second paragraph).

Reviewer #2 (Remarks to the Author):

1/ Major:

The mechanism by which cisplatin resistant cells reprogram fatty acid uptake and metabolism is not fully elucidated. It could be a nice addition to investigate whether cisplatin induces significant rewiring of gene expression in ovarian cancer cells and thereby determine the observed changes in lipid metabolism.

Response: We agree with the reviewer that elucidating the fatty acid metabolic reprogramming/rewiring is highly desired. Therefore, we performed bioinformatic analyses of RNA-seq datasets comparing OVCAR5 vs. OVCA5-cisR and SKOV3 vs. SKOV3-cisR (GSE 148003). We identified 52 genes in the Hallmark Fatty Acid Metabolism gene set that showed the same directionality in OVCAR5 and SKOV3 cell pairs (Supplementary Table 4). Then we generated heatmaps of hierarchical clustering of these overlapping genes for both cell pairs following the same gene order. The heatmaps showed distinct separation between sensitive and resistant cells. In both cell lines, we found that most of the FAO related genes (highlighted in red font) including CRAT, PPARA, ACOT8, HSD17B10, ACADVL, ACOX1 and DECR1 were upregulated in resistant cells, while fatty acid synthesis related genes (highlighted in green font) such as ME1, NSDHL, DHCR24, FASN, ELOVL5, ALDH3A2, ACSL4 and SERINC1 were downregulated (revised **Supplementary Fig.7a and b, and Supplementary Table 4**). These results together with expression analysis of fatty acid transporter genes (see below), support that fatty acid uptake and oxidation are both upregulated in the resistant cell lines compared to the sensitive cells.

We added this new analysis to the manuscript as supporting evidence. See page 15 third paragraph.

2/ Minor:

The authors mostly discuss fatty acid oxidation, but enhanced uptake of fatty acids might also be required for tumor growth (contributing to membrane biogenesis). If

relevant, I think this should also be discussed.

It could also be interesting to investigate/discuss whether reduced lipogenesis influences fatty acid beta-oxidation through malonyl-coa mediated inhibition of mitochondrial CPT?

Response: The reviewer has raised a good point. It is highly possible that enhanced fatty acid uptake may contribute to tumor growth by other mechanisms, for example, replenish the free fatty acid pool, which is likely reduced in resistant cells due to decreased lipogenesis. We have extended the discussion in the discussion section (page 21, first paragraph).

Indeed, malonyl-coa, a product of the rate limiting step in de novo lipogenesis, would decrease as lipogenesis reduced, and thus its allosteric inhibition on CPT1 would be released, resulting an increased FAO^{6,7}. We have extended related discussion in page 21, first paragraph.

Reviewer #3 (Remarks to the Author):

The manuscript titled “Metabolic Reprogramming from Glycolysis to Fatty Acid Uptake and beta-Oxidation in Platinum-Resistant Cancer Cells” by Junjie Li and colleagues. The authors describe the mechanism of metabolic programming in platinum-resistant ovarian cancer cells. They report that cisplatin-resistant cells exhibit increased uptake of exogenous fatty acids, decreased glucose uptake and de novo lipogenesis, indicating a metabolic switch from glycolysis-dependent to fatty acid uptake/ beta-oxidation dependent anabolic and energy metabolism. The authors used high throughput stimulated Raman scattering imaging and single cell analysis metabolic to show the changes of the uptake of glucose and fatty acids in paired sensitive, and resistant ovarian cancer cells and developed a metabolic index incorporating measurements of glucose derived anabolism and fatty acid uptake and found that this index linearly correlates with cisplatin-resistance in established ovarian cancer cell lines and in primary cells isolated from ovarian cancer patients. They also performed mechanistic studies to prove that the increased fatty acid uptake facilitates cancer cell survival under cisplatin-induced oxidative stress by enhancing energy production through beta-oxidation.

The hypothesis is clear, and the experimental approach used to test the hypothesis is logic and clear. Results are clearly presented with supporting supplementary data. Overall the manuscript is well written but the material and method section needs extensive editing and clarification.

Response: We appreciate the reviewer’s positive comments. We have carefully revised the Materials and Methods section by including more details and extensive editing.

Reviewer #4 (Remarks to the Author):

1. While the authors indicate that they are studying development of therapeutic chemoresistance, it is unclear if the problem addressed is being of inherent chemoresistance or acquired chemoresistance. Almost all the cell line models used are of artificially induced chemoresistance. The patient cells used have not been described well enough to know if these are from naïve patients or obtained from tumors that were exposed to chemotherapy. This may reveal if the FA dependency is adaptive metabolic alteration to development of resistance. In addition, for several years' carboplatin use has become the norm instead of cisplatin.

Response: We agree with the reviewer that the cell line models used here likely represent a metabolic adaptation during the development of acquired platinum resistance. Regarding the human specimens analyzed, we include additional clinical details (revised **Supplementary Table 2**). In brief, tumor specimens were obtained at the time of cytoreductive surgery either upfront or after neoadjuvant chemotherapy (interval debulking surgery). Patients were followed prospectively and received standard platinum-taxane chemotherapy. Platinum resistance was defined as disease recurring within 6 months from completing carboplatin-based chemotherapy, as assessed clinically, by CA125 criteria or CT scans. Given the limited number of samples available in this study, we are not able to determine if there is a difference in FA dependency between inherent and acquired resistant cells. Further studies using larger number of samples will be needed to address this question.

We have included this information to the revised manuscript (page 11, second paragraph).

2. It will be more relevant to create resistance *in vivo* by isolating cells after cisplatin treatment rather than the artificial *in vitro* dose incremental system. To demonstrate the true potential of SRS imaging to detect FAO at single cell level, these tumors can be used to show the heterogeneity of chemoresistance within the same tumor (Fig 1 demonstrated in tumors growth in mice +/- cisplatin). A confounding thought is that since post-chemo/recurrent tumor surgeries are not common, obtaining cancer cells to detect chemoresistance may be a hurdle to apply SRS imaging in tumor cells, and may have more applicability in inherent chemo resistant tumors.

Response: We fully agree with the reviewer that *in vivo* data from tumor models would be more compelling. To address this issue, we include new data from xenografts residual after treatment with carboplatin (resistant) or saline (sensitive) for 3 weeks. As anticipated, the tumor volume was significantly decreased in the carboplatin treated group compared to the saline treated group (revised **Supplementary Fig. 1h**). Ovarian cancer cells isolated from carboplatin treated tumors were confirmed to be less sensitive to carboplatin in a subsequent *in vitro* viability assay, compared to cells derived from the saline treated group (revised **Supplementary Fig. 1i**), supporting the separation as platinum resistant (post carboplatin treatment in mice) and platinum sensitive (control treated xenografts). Harvested tumor tissues were flash frozen and

sectioned for SRS imaging. We detected significantly higher lipid content in the xenografts residual after carboplatin treatment compared with control treated xenografts (revised **Fig. 1g** and **h**). The data support that the lipid content increases in resistant tumors in *in vivo* systems. Furthermore, lipid droplet distribution in single cells varied substantially in the carboplatin-treated tumor tissues, suggesting heterogeneity of chemoresistance within the same tumor.

Regarding the reviewer's point on SRS imaging detection for inherent vs. acquired tumor resistance, we agree that inherent chemo resistant tumors are logistically easier to be captured. We have included discussion in the revised manuscript (page 21, second paragraph).

3. To demonstrate that the increased lipid content of the chemo resistant cells is independent of de novo lipogenesis, demonstration that inhibition of de novo lipogenesis still results in increased lipid content/droplets in the resistant cells should be shown. Fig. 2 should be supported by functional measurements of glycolysis, mitochondrial oxidation, and FAO (probably by Seahorse) when the cells are grown in presence of glucose or exogenous FAs. Is the uptake of glucose vs FA dependent on the expression of transporters? Does GLUT1 expression gets decreased in resistant cells and in response to cisplatin while the FA transporters increase?

Response: In response, we performed additional experiments. We used SRS imaging and phasor analysis to measure lipid content in SKOV3 and SKOV3-cisR cell treated with an inhibitor of lipogenesis (FASN inhibitor C-75). Even when lipogenesis was inhibited by C-75, the lipid content in resistant cell SKOV3-cisR remained higher than cisplatin-sensitive SKOV3 (revised **Supplementary Fig.2c** and **d**), suggesting that the enhanced lipid content in cisplatin resistant cell is independent of de novo lipogenesis.

Furthermore, GLUT1 expression in SKOV3 and -cisR was measured by qPCR. We observed that GLUT1 expression in sensitive cells was higher than in resistant cells (revised **Supplementary Fig.4j**). Cisplatin treatment did not trigger a significant reduction of GLUT1 expression in OVCAR5 cells (revised **Supplementary Fig.4j**), but significantly increased expression of key FA transporter genes (FABP5 and FABP(PM)) (revised **Fig. 4f**), implying that GLUT1 expression downregulation is an adaptive change in cisplatin-resistant cells instead of an acute response to cisplatin treatment. These data are consistent with our observations regarding decreased glucose anabolism in resistant cells.

We have added these new data to the revised manuscript (page 9, second paragraph; page 13, first paragraph).

4. The applicability of the "metabolic index" is not clear. The *in vitro* created systems provide a very clear-cut correlation which may not apply to a heterogenous tumor. If the index score =0.5, how will the status of resistance be inferred?

Response: The reviewer raised an excellent question. As a proof of concept, based on the human patient data we have, a clear separation between cisplatin sensitive versus resistant ovarian cell metabolic index is presented in histogram (revised **Supplementary Fig.3f**). Receiver operating characteristic (ROC) yields a threshold value at 0.412 with high sensitivity of 1 specificity of 1 and AUC (area under curve) of 1, suggesting that this metabolic index has a high chance to successfully distinguish cisplatin sensitive and resistant ovarian cancer cells (revised **Supplementary Fig.3g**). This value needs to be validated using a larger sample size. We have added the new analysis to the revised manuscript (page 11, second paragraph).

5. The patient data (Fig. 3) should be supported by functional data showing increased FAO occurring in sensitive vs resistance cells.

Response: The suggestion to have additional functional data in patient cells besides metabolic imaging is important. However, this experiment is challenging due to the limited number of primary cells that can be isolated and the difficulty amplifying primary cells in culture to perform Seahorse measurement of FAO activity in these patient cells. This challenge can be overcome by using our optical imaging, which is applicable to small numbers of cells.

6. Limiting exogenous FAs resulted in decreased lipid accumulation. In these conditions, was sufficient glucose provided? Did the cells adapt by intaking more glucose? Will providing alternative fuels (glucose/glutamine etc.) make the cells overcome the FA deficiency? This is important to address as many chemo resistant cells, including ovarian have been reported to possess a flexible metabolism allowing them to utilize different energy pathways in times of stress to survive.

Response: In our experiment, high concentration of glucose (25 mM) and glutamine (2 mM) was provided in the medium. To verify if ovarian cancer cells would use glucose as alternative fuel in a fatty acid depleted environment, glucose uptake in cells cultured with control or reduced lipid content (10% FBS or 10% delipid serum) was evaluated (revised **Supplementary Fig.4e** and **f**). We observed that glucose uptake remained similar in both SKOV3 and -cisR when lipid amount in the culture environment was reduced, suggesting that glucose is not an alternative fuel for ovarian cancer cell in a fatty acid deficient environment. We added this new observation to the revised manuscript (page 12, first paragraph)

7. It is unclear why the expression of FA transporters was examined in only the OVCAR5 cell line set and not in all the various cell line sets. It would be more relevant to show the expression profiles in PEO1/4 cell lines and in the patient derived cell lines used in Fig. 3. The use of different cell lines to showcase different data throughout the manuscript is bothersome.

Response: We appreciate the comment. We performed gene expression analysis for FABP5 in PEO1/4 cell lines, shown in **supplementary Fig. 4h**. Our data show that resistant cell PEO4 expresses higher levels of the fatty acid transporter FABP5. Due to

limited number of primary cells isolated and difficulty in amplifying the primary cells in culture, we were unable to perform further gene analysis on patient derived cells used in Fig 3.

8. In Fig. 5, it is unclear why Seahorse FAO assay was not used to demonstrate specificity of FA induced OCR and its inhibition by etomoxir, as the authors clearly have access to the instrument. In the same experiment, rescue experiments with increased exogenous FAs and glucose should be presented to demonstrate the specificity of FAO and sensitivity.

Response: Following the reviewer's suggestion, we performed Seahorse FAO assay (OCR measurement in cells treated with FAO inhibitor etomoxir and with mitochondrial respiration inhibitors oligomycin, FCCP, rotenone and antimycin). Our new data demonstrate that etomoxir induced reduction in basal respiration, ATP production and maximal respiration is significantly higher in resistant cell OVCAR5-cisR than that in sensitive cell OVCAR5 suggesting an upregulated FAO in resistant cell (revised Fig. 5d and e). We have included the updated data in the revised manuscript (page 14, second paragraph). Furthermore, introducing exogenous FA leads to an increased reduction of OCR in response to etomoxir treatment, demonstrating higher FAO was induced by exogenous FA supporting the specificity of FAO measurement by Seahorse (Fig. R1).

Fig. R1 Seahorse measured OCR profiles of OVCAR5 and -cisR cells treated with or without etomoxir rescued by addition of exogenous fatty acid followed by injections of mitochondrial respiration inhibitors oligomycin, FCCP, rotenone and antimycin A.

9. The PDX experiment is well performed but may benefit by including additional PDX or xenograft models and the lipid accumulation demonstrated during the development of resistance. Including the FA transporter inhibitor BMS-309403 may strengthen the role of exogenous FA uptake, as omentum is known to be a source of exogenous FAs in ovarian cancer.

Response: We appreciate the reviewer's suggestion. To further confirm the lipid accumulation in platinum resistant tumors in vivo, we have included another OVCAR5 cell line derived xenograft model treated with or without carboplatin as platinum resistant or sensitive tumors. We then performed SRS imaging of lipids in tumor tissue with or without resistance. The results shown in revised Fig. 1g and h indicates heterogeneous lipid accumulation in resistant tumor tissues.

Regarding the role of FA transporter inhibitor BMS-309403, the Lengyel group has shown it reduces carboplatin resistance in vitro and synergistically suppresses tumor metastasis with carboplatin in xenograft model ⁸. Together, these data show lipid accumulation and an important role of fatty acid uptake in ovarian cancer.

10. Can the authors address the role of nutrient status in the tumor microenvironment. Recent work suggests that limiting nutrient (for example glucose or glutamine) may force the cancer cell to start using alternative fuel sources. Since the authors propose the affinity of chemo resistant cells towards FA over glucose, this should be discussed in context of ovarian cancer.

Response: We appreciate the reviewer's suggestion. We have included a discussion of this topic in the discussion section (**page 18, first paragraph**).

In Summary, we performed an extensive revision of the manuscript including substantial new experimentation and new results to address all of the concerns raised by the reviewers. We look forward to your positive evaluation of this work.

Reference

1. Shen, Y. *et al.* Metabolic activity induces membrane phase separation in endoplasmic reticulum. *Proc Natl Acad Sci U S A* **114**, 13394-13399 (2017).
2. Zhang, X. *et al.* Celastrol Reverses Palmitic Acid-Induced Insulin Resistance in HepG2 Cells via Restoring the miR-223 and GLUT4 Pathway. *Can J Diabetes* **43**, 165-172 (2019).
3. Gonzalez-Giraldo, Y., Garcia-Segura, L.M., Echeverria, V. & Barreto, G.E. Tibolone Preserves Mitochondrial Functionality and Cell Morphology in Astrocytic Cells Treated with Palmitic Acid. *Mol Neurobiol* **55**, 4453-4462 (2018).
4. Lin, L. *et al.* Functional lipidomics: Palmitic acid impairs hepatocellular carcinoma development by modulating membrane fluidity and glucose metabolism. *Hepatology* **66**, 432-448 (2017).
5. Zeng, L. *et al.* Saturated fatty acids modulate cell response to DNA damage: implication for their role in tumorigenesis. *PLoS One* **3**, e2329 (2008).
6. Ma, Y. *et al.* Fatty acid oxidation: An emerging facet of metabolic transformation in cancer. *Cancer Lett.* **435**, 92-100 (2018).
7. Melone, M.A.B. *et al.* The carnitine system and cancer metabolic plasticity. *Cell Death Dis.* **9**, 228 (2018).
8. Mukherjee, A. *et al.* Adipocyte-Induced FABP4 Expression in Ovarian Cancer Cells Promotes Metastasis and Mediates Carboplatin Resistance. *Cancer Res* **80**, 1748-1761 (2020).

REVIEWERS' COMMENTS

Reviewer #1 (Remarks to the Author):

The authors have properly addressed all my questions. I hence support its publication in Nat. Commun.

Reviewer #2 (Remarks to the Author):

My comments have been addressed.

Reviewer #4 (Remarks to the Author):

The authors have addressed all the concerns very well.

Congratulations on the great work.

I have no further comments.